



# N₂ fixation in the Mediterranean Sea related to the composition of the diazotrophic community, and impact of dust under present and future environmental conditions

Céline Ridame[1], Julie Dinasquet[2,3], Søren Hallstrøm[4], Estelle Bigeard[5], Lasse Riemann[4], France Van Wambeke[6], Matthieu Bressac[7], Elvira Pulido-Villena[6], Vincent Taillandier[7], Frederic Gazeau[7], Antonio Tovar-Sanchez[8], Anne-Claire Baudoux[5], Cécile Guieu[7]

[1] Sorbonne University, CNRS, IRD, LOCEAN: Laboratoire d'Océanographie et du Climat: Expérimentation et Approches Numériques, UMR 7159, 75252 Paris Cedex 05, France
[2] Scripps Institution of Oceanography, University of California San Diego, USA
[3] Sorbonne University, CNRS, Laboratoire d'Océanographie Microbienne, LOMIC, 66650 Banyuls-sur-Mer, France
[4] Marine Biology Section, Department of Biology, University of Copenhagen, 3000 Helsingør, Denmark
[5] Sorbonne University, CNRS, Station Biologique de Roscoff, UMR 7144 Adaptation et Diversité en Milieu Marin, France
[6] Aix-Marseille Université, Université de Toulon, CNRS/INSU, IRD, Mediterranean Institute of Oceanography (MIO), UM 110, 13288, Marseille, France
[7] Sorbonne Université, CNRS, Laboratoire d'Océanographie de Villefranche, LOV, 06230 Villefranche-sur-Mer, France
[8] Department of Ecology and Coastal Management, Institute of Marine Sciences of Andalusia (CSIC), 11510 Puerto Real, Cádiz, Spain

*Correspondence to*: Céline Ridame (celine.ridame@locean.ipsl.fr)

**Abstract.** N₂ fixation rates were measured in the 0-1000 m layer at 13 stations located in the open western and central Mediterranean Sea (MS) during the PEACETIME cruise (late spring 2017). While the spatial variability of N₂ fixation was not related to Fe, P nor N stocks, the surface composition of the diazotrophic community indicated a strong eastward increasing longitudinal gradient for the relative abundance of non-cyanobacterial diazotrophs (NCD) (mainly γ-Proteobacteria) and conversely eastward decreasing for UCYN-A (mainly -A1 and -A3) as did N₂ fixation rates. UCYN-A4 and -A3 were identified for the first time in the MS. The westernmost station influenced by Atlantic waters, and characterized by highest stocks of N and P, displayed a patchy distribution of diazotrophic activity with an exceptionally high rate in the euphotic layer of 72.1 nmol N L$^{-1}$ d$^{-1}$, which could support up to 19 % of primary production. At this station at 1%PAR depth, UCYN-A4 represented up to 94 % of the diazotrophic community. These *in situ* observations of higher UCYN-A relative abundance in nutrient rich stations while NCD increased in the more oligotrophic stations, suggest that the nutrient conditions could determine the composition of the diazotrophic communities and in turn the N₂ fixation rates.

The impact of Saharan dust deposition on N₂ fixation and diazotrophic communities was also investigated, under present and future projected conditions of temperature and pH during short term (3-4 days) experiments at three stations. New nutrients





from simulated dust deposition triggered a significant stimulation of N$_2$ fixation (from 41 % to 565 %). The strongest increase in N$_2$ fixation was observed at the stations dominated by NCD and did not lead on this short time scale to change in the diazotrophic community composition. Under projected future conditions, N$_2$ fixation was either exacerbated or unchanged; in that later case this was probably due to a too low nutrient bioavailability or an increased grazing pressure. The future warming and acidification likely benefited NCD (*Pseudomonas)* and UCYN-A2 while disadvantaged UCYN-A3

without knowing which effect (alone or in combination) is the driver, especially since we do not know the temperature optima of these species not yet cultivated as well as the effect of acidification.

## 1. Introduction

The Mediterranean Sea (MS) is considered as one of the most oligotrophic regions of the world's ocean (Krom et al., 2004; Bosc et al., 2004). It is characterized by a longitudinal gradient in nutrient availability, phytoplanktonic biomass and primary

production (PP) decreasing eastward (Manca et al., 2004; D'Ortenzio and Ribera d'Alcalà, 2009; Ignatiades et al., 2009; Siokou-Frangou et al., 2010; El Hourany et al., 2019). From May to October, the upper water column is well-stratified (D'Ortenzio et al., 2005), and the sea surface mixed layer (SML) becomes nutrient-depleted leading to low PP (e.g. Lazzari et al., 2012). Most measurements of N$_2$ fixation during the stratified period have shown low rates ($\leq$ 0.5 nmol N L$^{-1}$ d$^{-1}$) in surface waters of the open MS (Ibello et al., 2010; Bonnet et al., 2011; Yogev et al., 2011; Ridame et al., 2011; Rahav et al.,

2013a; Benavides et al., 2016) indicating that N$_2$ fixation represents a minor source of bioavailable nitrogen in the MS (Krom et al., 2010; Bonnet et al., 2011). These low rates are likely related to the extremely low bioavailability in dissolved inorganic phosphorus (DIP) (Rees et al., 2006; Ridame et al., 2011). The high concentrations of dissolved iron (DFe) in the SML due to accumulated atmospheric Fe deposition (Bonnet and Guieu 2006; Tovar-Sánchez et al. 2020; Bressac et al., 2021), suggest that the bioavailability of Fe is not a controlling factor of N$_2$ fixation (Ridame et al., 2011). Occasionally,

high N$_2$ fixation rates have been reported locally in the northwestern (17 nmol N L$^{-1}$ d$^{-1}$; Garcia et al., 2006) and eastern MS (129 nmol N L$^{-1}$ d$^{-1}$; Rees et al., 2006). Usually, the low N$_2$ fixation rates in the Mediterranean offshore waters are associated with low abundance of diazotrophs, mainly dominated by unicellular organisms (Man-Aharonovich et al., 2007; Yogev et al., 2011; Le Moal et al., 2011). Unicellular diazotrophs from the photo-heterotrophic group A (UCYN-A, Zehr et al., 1998) largely dominated the cyanobacteria assemblage in the MS (Le Moal et al., 2011), and very low concentrations of

filamentous diazotrophic cyanobacteria have only been recorded in the eastern basin (Bar-Zeev et al., 2008; Le Moal et al., 2011; Yogev et al., 2011). The UCYN-A cluster consist of four sublineages: UCYN-A1, -A2, -A3 and -A4 (Thompson et al., 2014; Farnelid et al., 2016; Turk Kubo et al., 2017; Cornejo-Castillo et al., 2019), of which only UCYN-A1 and -A2 have been previously detected in the MS (Man-Aharonovich et al., 2007; Martinez-Perez et al., 2016; Pierrela Karlusich et al., 2021). Heterotrophic diazotrophs are widely distributed over the offshore surface waters (Le Moal et al, 2011), and the

decreasing eastward gradient of surface N$_2$ fixation rate could be related to a predominance of photo-autotrophic diazotrophs in the western basin and a predominance of heterotrophic diazotrophs in the eastern one (Rahav et al. 2013a).



The MS is strongly impacted by periodic dust events, originating from the Sahara, which have been recognized as a significant source of macro- and micronutrients, to the nutrient depleted SML during stratified periods (Guieu and Ridame, 2020 and references therein; Mas et al., 2020). Results from Saharan dust seeding experiments during open sea microcosms

and coastal mesocosms in the MS, showed stimulation of both PP (Herut et al., 2005; Ternon et al., 2011; Ridame et al., 2014 ; Herut et al., 2016) and heterotrophic bacterial production (BP) (Pulido-Villena et al., 2008, 2014; Lekunberri et al., 2010; Herut et al., 2016). Experimental Saharan dust seeding was also shown to enhanced $N_2$ fixation in the western and eastern MS (Ridame et al., 2011; Ternon et al., 2011; Ridame et al., 2013; Rahav et al., 2016a) and to alter the composition of the diazotrophic community (Rahav et al., 2016a), as also shown in the tropical North Atlantic (Langlois et al., 2012).

The MS has been identified as one of the primary hot-spots for climate change (Giorgi, 2006). Future sea surface warming and associated increase in stratification (Somot et al., 2008) might reinforce the importance of atmospheric inputs as a source of new nutrients for biological activities, and in particular diazotrophic microorganisms during the stratified period. This fertilizing effect could also be enhanced by the expected decline in pH (Mermex Group, 2011), which could increase the nutrient dust solubility in seawater. Under nutrients repleted conditions, predicted elevated temperature and $CO_2$

concentration favor the growth and $N_2$ fixation of the filamentous cyanobacteria *Trichodesmium* and of the photo-autotrophic UCYN-B and -C (Webb et al., 2008; Hutchins et al., 2013; Fu et al., 2008, 2014; Eichner et al., 2014; Jiang et al., 2018), whereas effects on UCYN-A and non-cyanobacterial diazotrophs (NCD) are uncertain.

In this context, the first objective of this study is to investigate during the season characterized by strong stratification and low productivity, the spatial variability of $N_2$ fixation rates in relation to nutrients availability and diazotrophic communities

composition. The second objective was to study, for the first time, the impact of a realistic Saharan deposition event in the open MS, on $N_2$ fixation rates and diazotrophic communities composition under present and realistic projected conditions of temperature and pH for 2100.

## 2. Materials and Methods

### 2.1 Oceanographic cruise

All data were acquired during the PEACETIME cruise (ProcEss studies at the Air-sEa Interface after dust deposition in the MEditerranean sea) in the western and central MS on board the R/V *Pourquoi Pas ?* from May 10 to June 11, 2017 (http://peacetime-project.org/) (see the detailed description in Guieu et al., 2020). The cruise track including ten short stations (ST1 to ST10) and three long stations (TYR, ION and FAST) is shown in Fig.1 (coordinates in Table S1). Stations 1

and 2 were located in the Liguro-Provencal basin; stations 5, 6, and TYR, in the Tyrrhenian Sea; stations 7, 8, and ION in the Ionian Sea; and stations 3, 4, 9, 10 and FAST in the Algerian basin.

### 2.2 Dust seeding experiments

Experimental dust seedings into six large tanks were conducted at each of the three long stations (TYR, ION and FAST),

under present and future conditions of temperature and pH. These stations were chosen for their differences in oligotrophic



conditions and metabolic activity of the diazotrophs. The experimental setup is fully described in a companion paper (Gazeau et al., 2021a). Briefly, six climate reactors (volume of about 300 L) made in high density polyethylene were placed in a temperature-controlled container, and covered with a lid equipped with LEDs to reproduce natural light cycle. The tanks were filled with unfiltered surface seawater collected at ~5m with a peristaltic pump at the end of the day (T-12h) before the

start of the experiments the next morning (T0). Two replicate tanks were amended with mineral Saharan dust (Dust treatments D1 and D2) simulating a high but realistic atmospheric dust deposition of 10 g m$^{-2}$ (Guieu et al., 2010b). Two other tanks were also amended with Saharan dust (same dust flux as in the Dust treatment) under warmer (~ +3° C) and more acidic water conditions (~ -0.3 pH unit) (Greenhouse treatments G1 and G2). This corresponds to the IPCC projections for 2100 under RCP8.5 (IPCC 2019). Seawater in G1 and G2 was warmed overnight to reach +3° C and acidified through the

addition of $CO_2$-saturated 0.2 μm-filtered seawater (~1.5 L in 300 L). The difference in temperature between G (Greenhouse) tanks and other tanks (C, Controls and D, Dust) was +3° C, +3.2° C and +3.6° C at TYR, ION and FAST, respectively, and the decrease in pH was -0.31, -0.29 and -0.33 at TYR, ION and FAST, respectively (Gazeau et al., 2021a). Two tanks were filled with untreated water (Controls C1 and C2). The experiment at TYR and ION lasted three days while the experiment at FAST lasted four days. The sampling session took place every morning at the same time over the duration

of the experiments.

The fine fraction (< 20 μm) of a Saharan soil collected in southern Tunisia used in this study, has been previously used for the seeding of mesocosms in the frame of the DUNE project (a DUst experiment in a low-Nutrient, low-chlorophyll Ecosystem). Briefly, the dust was previously subjected to physico-chemical transformations mimicking the mixing between dust and pollution air masses during atmospheric transport (Desboeufs et al, 2001; Guieu et al., 2010b). This dust contained

0.055 ± 0.003 % of P, 1.36 ± 0.09 % of N, and 2.26 ± 0.03 % of Fe, in weight (Desboeufs et al., 2014). Right before the artificial seeding, the dust was mixed with 2 L of ultrapure water in order to mimic a wet deposition event and sprayed at the surface of the climate reactors D and G.

**2.3 N$_2$ fixation and primary production**

All materials were acid washed (HCl Suprapur) following trace metal clean procedures. Before sampling, bottles were rinsed three times with the sampled seawater. For the *in situ* measurements, seawater was sampled using a trace metal clean (TMC) rosette equipped with 24 GO-FLO Bottles (Guieu et al., 2020). At each station, 7 to 9 depths were sampled between surface and 1000 m for N$_2$ fixation measurements, and 5 depths between surface and ~100 m for primary production measurements. During the seeding experiments, the six tanks were sampled for simultaneous determination of N$_2$- and CO$_2$ net fixation rates

before dust seeding (initial time T0) and one day (T1), two days (T2), and three days (T3) after dust addition at TYR and ION stations. At FAST, the last sampling took place four days (T4) after dust addition.

Net N$_2$ fixation rates were determined using the $^{15}$N$_2$ gas-tracer addition method (Montoya et al., 1996), and net primary production using the $^{13}$C-tracer addition method (Hama et al., 1983). Immediately after sampling, 1 mL of NaH$^{13}$CO$_3$ (99 %, Eurisotop) and 2.5 ml of 99 % $^{15}$N$_2$ (Eurisotop) were introduced to 2.3 L polycarbonate bottles through a butyl septum for





simultaneous determination of $N_2$- and $CO_2$-fixation. $^{15}N_2$ and $^{13}C$ tracers were added to obtain a ~10 % final enrichment. Then, each bottle was vigorously shaken before incubation for 24 h. The *in situ* samples from the euphotic zone were incubated in on-deck containers with circulating seawater, equipped with blue filters (percentages of attenuation: 70, 52, 38, 25, 14, 7, 4, 2 and 1 %) to simulate *in situ* irradiance at different depths. Samples for $N_2$ fixation determination in the aphotic layer were incubated in the dark in thermostated incubators set at *in situ* temperature. *In situ* $^{13}C$-PP will not be discussed in

this paper as $^{14}C$-PP rates are presented in Maranon et al. (2021). The *in situ* $^{13}C$-PP were used in the present study to estimate the contribution of $N_2$ fixation to PP.

Samples from the dust addition experiments were incubated in two tanks dedicated to incubation: one tank at the same temperature and irradiance as tanks C and D, and another one at the same temperature and irradiance as tanks G. It should be noted that $^{14}C$-PP was also measured during the seedings experiments (Gazeau et al., 2021b).

After 24 h incubation, 2.3 L were filtered onto pre-combusted 25 mm GF/F filters, and filters were stored at −25° C. Filters were then dried at 40° C for 48 h before analysis. Particulate carbon (C) and nitrogen (N) as well as $^{15}N$ and $^{13}C$ isotopic ratios were quantified using an online continuous flow elemental analyzer (Flash 2000 HT), coupled with an Isotopic Ratio Mass Spectrometer (Delta V Advantage via a conflow IV interface from Thermo Fischer Scientific). $N_2$ fixation rates were calculated by isotope mass balanced as described by Montoya et al. (1996). The detection limit for $N_2$ fixation, calculated

from significant enrichment and lowest particulate nitrogen was 0.04 nmol N $L^{-1}$ $d^{-1}$. From these measurements, the molar C:N ratio in the particulate matter was calculated and used to estimate the contribution of $N_2$ fixation to primary production. As a rough estimate of the potential impact of bioavailable N input from $N_2$ fixation on BP, we converted BP in N demand using the molar ratio C/N of 7.3 (Nagata et al., 1986). Trapezoidal method was used to calculate integrated rates over the SML, the euphotic layer (from surface to 1 % photosynthetically available radiation (PAR) depth) and the 0-1000 m water

column.

It must be noted that $N_2$ fixation rates measured by the $^{15}N_2$-tracer gas addition method may have been underestimated due to incomplete $^{15}N_2$ gas bubble equilibration (Mohr et al., 2010). However, this potential underestimation is strongly lowered during long incubation (24h).

The relative changes (RC, in %) in $N_2$ fixation in the dust experiments were calculated following :

$$RC \ (\%) = 100 \times \frac{(N_2 FIXATION_T - N_2 FIXATION_{Control}) Tx}{(N_2 FIXATION_{Control}) Tx}$$

with $N_2$ Fixation$_T$ is the rate in D1, D2, G1 or G2 at Tx, $N_2$ Fixation$_{Control}$ is the mean of the duplicated controls (C1 and C2) at Tx, and Tx the time of the sampling.

## 2.4 Composition of the diazotrophic community

Samples for characterization of the diazotrophic communities were collected during the dust seeding experiments in the six

tanks at initial time before seeding (T0) and final time (T3 at TYR and ION, and T4 at FAST). Three liters of water were collected in acid-washed containers from each tank, filtered onto 0.2 µm PES filters (Sterivex) and stored at -80° C until





DNA extraction. The composition of the diazotrophic community was also determined at four depths (10, 61, 88 and 200 m)
at station 10. Here, 2 L seawater were collected from the TMC rosette. Immediately after collection, seawater was filtered
under low vacuum pressure through a 0.2 µm-Nuclepore membrane and stored at -80° C in cryovials. Nucleic acids were
obtained from both filter types using phenol-chloroform extraction followed by purification (NucleoSpin® PlantII kit;
Macherey-Nagel). DNA extracts were used as templates for PCR amplification of the nifH gene by nested PCR protocol as
fully described in Bigeard et al., (2021, protocol.io). Following polymerase chain reactions, DNA amplicons were purified,
and quantified using NanoQuant Plate™ and Tecan Spark® (Tecan Trading AG, Switzerland). Each PCR product was
normalized to 30ng/µl in final 50µl and sent to Genotoul (https://www.genotoul.fr/, Toulouse, France) for high throughput
sequencing using paired-end 2x250bp Illumina MiSeq. All reads were processed using the Quantitative Insight Into
Microbial Ecology 2 pipeline (QIIME2 v2020.2, Bolyen et al., 2019). Reads were truncated to 350 bp based on sequencing
quality, denoised, merged and chimera-checked using DADA2 (Callahan et al., 2016). A total of 1,029,778 reads were
assigned to 635 amplicon sequence variants (ASVs). The table was rarefied by filtering at 1 % relative abundance per sample
cut-off that reduced the dataset to 97 ASVs accounting for 98.27 % of all reads. Filtering for homologous genes was done
using the NifMAP pipeline (Angel et al., 2018) and translation into amino acids using FrameBot (Wang et al., 2013). This
yielded 235 ASVs accounting for 1,022,184 reads (99 %). These remaining ASVs were classified with DIAMOND blastp
(Buchfink et al 2015) using a FrameBot translated nifH database (phylum level version; M. A. Moynihan. 2020) based on
the ARB database from the Zehr Lab (version June 2017; https://www.jzehrlab.com/nifh). NifH cluster and subcluster
designations were assigned according to (Frank et al., 2016). UCYN-A sublineages were assigned by comparison to UCYN-
A reference sequences (Farnelid et al., 2016; Turk-Kubo et al., 2017). All sequences associated with this study have been
deposited under the BioProject ID: PRJNA693966. Alpha and beta-diversity indices for community composition, were
estimated after randomized subsampling. Analyses were run in QIIME 2 and in Primer v.6 software package (Clarke and
Warwick, 2001).

**2.5 Complementary data from PEACETIME companions papers**

**Bacterial production-** Heterotrophic bacterial production (BP, *sensus stricto* referring to prokaryotic heterotrophic
production) was determined on board using the microcentrifuge method with the $^3$H- leucine ($^3$H-Leu) incorporation
technique to measure protein production (Smith and Azam, 1992). The detailed protocol and the rates of BP are presented in
Van Wambeke et al. (2021) for measurements in the water column, and in Gazeau et al. (2021b) for measurements over the
course of the dust seeding experiments.

**Dissolved Fe-** Dissolved iron (DFe) concentrations (< 0.2 µm) were measured by flow injection analysis with online
preconcentration and chemiluminescence detection (FIA-CL). The detection limit was 15 pM (Bressac et al., 2021). DFe
concentrations in the water column along the whole transect are presented in Bressac et al. (2021) and for the dust seeding
experiments in Roy-Barman et al. (2021).

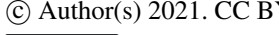



**Dissolved inorganic phosphorus and nitrate-** Concentrations of DIP and nitrate ($NO_3^-$) were analyzed immediately after collection on 0.2 μm filtered seawater using a segmented flow analyzer (AAIII HR Seal Analytical) according to Aminot and Kérouel (2007) with respective detection limits of 0.02 μmol $L^{-1}$ and 0.05 μmol $L^{-1}$. Samples with concentrations below the limit of detection with standard analysis were analyzed by spectrophotometry using a 2.5 m long waveguide capillary cell (LWCC) for DIP (Pulido-Villena et al., 2010) and a 1 m LWCC for $NO_3^-$ (Louis et al., 2015); the limit of detection was 1

nM for DIP and 6 nM for $NO_3^-$. Samples for determination of $NO_3^-$ at nanomolar level were lost from stations 1 to 4. The dust addition experiments data are detailed in Gazeau et al. (2021a). The water column data are fully discussed in Pulido-Villena et al. (2021) and Van Wambeke et al. (2021).

### 2.6 Statistical analysis

Pearson's correlation coefficient was used to test the statistical linear relationship ($p < 0.05$) between $N_2$ fixation and other variables (BP, PP, DFe, DIP, $NO_3^-$). In the dust seeding experiments, means at initial time (T0) before dust amendment (average at T0 in C and D treatments; n = 4, see Table 2) were compared using a one-way ANOVA followed by a Tukey means comparison test (α = 0.05). When assumptions for ANOVA were not respected, means were compared using a Kruskal–Wallis test and a post hoc Dunn test. To test significant differences ($p < 0.05$) between the slopes of $N_2$ fixation as a

function of time in the C, D and G treatments (n = 8), an Ancova was performed on data presenting a significant linear relationship with time (Pearson's correlation coefficient, $p < 0.05$). Statistical tests were done using the XLSTAT and R studio softwares.

### 3. Results


### 3.1 *In situ* $N_2$ fixation

### 3.1.1 Vertical and longitudinal distribution of $N_2$ fixation

Over the cruise, the water column was well stratified with a shallow SML varying from 7 to 21 m depth (Table S1). Detectable $N_2$ fixation rates in the 0-1000 m layer ranged from 0.04 to an exceptionally high rate of 72.1 nmol N $L^{-1}$ $d^{-1}$ at

station 10 (Fig.2). Vertical $N_2$ fixation profiles exhibited a similar shape at all stations with maximum values within the euphotic layer and undetectable values below 300 m depth (except at stations 1 and 10 with rates ~ 0.05 nmol N $L^{-1}$ $d^{-1}$ at 500 m depth). Within the euphotic layer, all the rates were well above the detection limit (DL = 0.04 nmol N $L^{-1}$ $d^{-1}$; minimum *in situ* $N_2$ fixation = 0.22 nmol N $L^{-1}$ $d^{-1}$). The highest rates were generally found below the SML and the lowest at the base of the euphotic layer or within the SML (Fig.2). The lowest $N_2$ fixation rates integrated over the euphotic and aphotic (defined

as 1 % PAR depth to 1000 m) layers were found at station 8, and the highest at station 10 (Table 1). On average, 59 ± 16 % of $N_2$ fixation (min 42 % at TYR and ION, max 97 % at station 10) took place within the euphotic layer (Table 1). The contribution of the SML integrated $N_2$ fixation to the euphotic layer integrated $N_2$ fixation was low, on average 17 ± 10 %.





Surface (~ 5 m) and euphotic layer integrated $N_2$ fixation rates exhibited a longitudinal gradient decreasing eastward (r = -0.61 and r = -0.60, p < 0.05, respectively) (Fig.3). Integrated $N_2$ fixation rates over the SML, aphotic and 0-1000 m layers

displayed no significant trend with longitude (p > 0.05). It should be noted that longitudinal trends with stronger correlations were observed for $^{13}C$-PP and BP (r = -0.81 and r = -0.82, p < 0.05, respectively, Fig.S1) as well as DIP and $NO_3^-$ stocks (r = -0.68 and r = -0.80, p < 0.05, no correlation with DFe stock; data not shown) integrated over the euphotic layer.

### 3.1.2 $N_2$ fixation and composition of diazotrophs at Station 10

The westernmost station 10 was in sharp contrast to all other stations with an euphotic integrated $N_2$ fixation on average 44 times higher (Table 1) due to high rates of 2.9 at 37 m and 72.1 nmol N $L^{-1}$ $d^{-1}$ at 61 m (i.e. at the deep chlorophyll-a maximum, DCM) (Fig.2). That rate at 61 m was associated with a maximum in PP but not with a maximum in BP. From surface to 200 m depth, the *nifH* community composition was largely dominated by ASVs related to different UCYN-A groups (Fig.4), that represented 86 % at 200 m and up to 99.5 % at the DCM. No UCYN-B and –C as filamentous

diazotrophs were detected. The relative abundance of NCD (mainly γ-Proteobacteria *Pseudomonas*) increased with depth (r = 0.96, p < 0.05) to reach about 8 % in the mesopelagic layer (200 m). UCYN-A1 and -A4 dominated the total diazotrophic community (from 51 to 99 %). All four UCYN-A had different vertical distributions: the relative abundances of UCYN-A1 and -A3 were the highest in surface while UCYN-A4 was dominant at the most productive depths (61 and 88 m). At 61 m depth, where the unusually high rate of $N_2$ fixation was detected, the community was dominated by both UCYN-A4 (58 %)

and UCYN-A1 (41 %).

### 3.1.3 $N_2$ fixation versus primary production, heterotrophic bacterial production, nutrients

For statistical analysis, $N_2$ fixation rates from station 10 were not included in order not to bias the analysis. $N_2$ fixation rate integrated over the euphotic layer correlated strongly with PP (r = 0.71, p < 0.05) and BP (r = 0.76; p < 0.05) (Fig.5).

Integrated $N_2$ fixation over the euphotic layer (and over the SML) was not correlated with the associated DFe, DIP and $NO_3^-$ stocks (p > 0.05). It should be noted that DIP and $NO_3^-$ stocks correlated positively with PP and BP (p < 0.05) over the euphotic layer (no correlation between DFe stock, and PP, BP).

### 3.2 Response of $N_2$ fixation and composition of the diazotrophic communities to dust seeding

### 3.2.1 Initial characteristics of the tested seawater

$N_2$ fixation and BP were the highest at FAST while PP was the highest at FAST and ION (Table 2). The $N_2$ fixation rates were similar at ION and TYR and significantly higher (factor ~2.6) at FAST. At TYR and ION, the diazotrophs community was largely dominated by NCD (on average 94.5 % of the total diazotrophic community) whereas at FAST, diazotrophic cyanobacteria, mainly UCYN-A, represented on average 91.4 % of the total diazotrophic community. $NO_3^-$ concentration

was the highest at FAST (59 nM) while DIP concentration was the highest at TYR (17 nM) and the lowest at ION (7 nM).





The molar $NO_3^-$/DIP ratio was strongly lower than the Redfield ratio (16/1) indicating a potential N limitation of the phytoplanktonic activity in all experiments. DFe concentrations were all higher than 1.5 nM.

### 3.2.2 Changes in $N_2$ fixation in response to dust seeding and relationship with changes in primary and heterotrophic

**bacterial production**

All the dust seedings led to a significant stimulation of $N_2$ fixation relative to the controls under present and future climate conditions (D and G treatments) (Figs.6, S2). The reproducibility between the replicated treatments was good at all stations (mean CV % < 14 %). The maximum $N_2$ fixation relative change (RC) was the highest at TYR (+434-503 % in D1 and D2, +478-565 % in G1 and G2) then at ION (+256-173 % in D1 and D2, and +261-217 % in G1 and G2) and finally at FAST

(+41-49 % in D1 and D2 and +97-120 % in G1 and G2) (Fig.7). At TYR and FAST, dust addition stimulated $N_2$ fixation more in the G treatment than in D, whereas at ION the response was similar between the treatments (Fig.S2). $N_2$ fixation measured during the dust seeding experiments correlated strongly with PP at FAST (r = 0.90, p < 0.05), and with BP at TYR and ION (r > 0.76, p < 0.05).

### 3.2.3 Changes in the diazotrophic composition in response to dust seeding

At TYR and ION, the diazotrophic communities before seeding were largely dominated by NCD (~ 94.5 % of total ASVs, Figs.8,S4, S5), mainly related to γ-proteobacteria (related to *Pseudomonas*). Some of these ASVs had low overall abundance, and therefore did not appear in the top 20 ASVs (Fig.8) but could nevertheless, account for up to 16 % in a specific sample. Filamentous cyanobacteria (*Katagnymene*) were also observed at both stations (~ 4.7 % of the total diazotrophs). At T0 at TYR and ION, the variability between replicates was high (C1T0 at TYR was removed due to poor

sequencing quality) while it was low at FAST. The community at FAST was initially dominated by UCYN-A phylotypes, mostly represented by UCYN-A1 and -A3 (relative abundance of UCYN-A1 and -A3: 34 ± 6 % and 45 ± 2 % of the total diazotrophic composition, respectively) (Fig.8). For ION and FAST experiments, *Pseudomonas* related ASVs were more abundant in G treatments at T0 relative to Control and Dust treatments (T0). At the end of the TYR and ION experiments, the community from all treatments appeared to converge (Fig.S4) due to the increase of a few γ-proteobacteria (mainly

*Pseudomonas)* that strongly increased in all treatments (Fig.8). At FAST, no difference in the relative abundances of diazotrophs was recorded between D treatment and the controls at T4 whereas the contribution of NCD was higher (82 % in G vs. 63 % in D) and that of UCYN-A was lower (13 % in G vs. 31 % in D) in G treatment relative to D, at T4.

### 4. Discussion

Late spring, at the time of sampling, all the stations were well-stratified and characterized by oligotrophic conditions increasing eastward (Maranon et al., 2021; Fig.8 in Guieu et al., 2020). $NO_3^-$ and DIP concentrations were low in the SML, from 9 to 135 nM for $NO_3^-$ (Van Wambeke et al., 2021) and from 4 to 17 nM for DIP (Pulido-Villena et al., 2021); the highest stocks were measured at the westernmost station (St 10) (Table S1).





### 4.1 General features in N$_2$ fixation and diazotroph community composition

N$_2$ fixation rates in the aphotic layer were in the range of those previously measured in the western open MS (Benavides et al., 2016) and accounted, on average, for 41 % of N$_2$ fixation in the 0-1000 m layer, suggesting that a large part of the total diazotrophic activity was related to heterotrophic NCD in the aphotic layer. N$_2$ fixation rates in the euphotic layer were of the same order of magnitude (data from St10 excluded) than those previously measured in the open MS in spring and summer

(Bonnet et al., 2011; Rahav et al., 2013a). At the tested stations, the surface diazotrophic cyanobacteria were largely dominated by UCYN-A (~ 93 % of the total diazotrophic cyanobacteria, mostly UCYN-A1 and -A3) and the NCD community by γ-proteobacteria (~ 95 % of the total NCD). This is the first time that UCYN-A3 and -A4 are detected in the MS. The photo-autotrophic N$_2$ fixation was negligible as no UCYN-B and -C were detected and very low abundance of filamentous cyanobacteria was observed.


### 4.2 Longitudinal gradient of N$_2$ fixation related to the composition of the diazotrophic communities

At station 10 and FAST, the surface diazotrophic communities were largely dominated by UCYN-A (> 91 %) whereas at TYR and ION they were dominated by NCD (> 94 %) which highlights the predominance of photo-heterotrophic diazotrophy in the western waters of the Algerian Basin and of NCD-supported diazotrophy in the Tyrrhenian and Ionian

basins. Surface N$_2$ fixation exhibited a longitudinal gradient decreasing eastward as previously reported (Bonnet et al., 2011, Rahav et al., 2013a). Strong longitudinal gradients decreasing eastward for the relative abundance of UCYN-A ($r = -0.93$, $p < 0.05$) and inversely increasing eastward for NCD were observed ($r = 0.89$, $p < 0.05$) (Fig.9). Despite no quantitative abundances of distinct diazotrophs for the studied area (this and previously published studies), the intensity of the bulk N$_2$ fixation rate was likely related to the overall composition of the diazotrophic communities (here relative abundance of

UCYN-A versus NCD). Indeed, surface N$_2$ fixation rates correlated positively with the relative abundance of UCYN-A (mainly A1 and A3) ($r = 0.98$, $p < 0.05$) and negatively with the relative abundance of NCD ($r = -0.94$, $p < 0.05$). This could be related, in part, to the variability of the cell-specific N$_2$ fixation rates that were shown to be higher for UCYN-A relative to NCD (Turk-Kubo et al., 2014; Bentzon-Tilia et al., 2015; Martinez-Perez et al., 2016; Pearl et al., 2018; Mills et al., 2020). Besides, in Atlantic and Pacific Ocean areas when the diazotrophic community is dominated by unicellular

organisms, high N$_2$ fixation rates are mostly associated with a predominance of UCYN-A, and low rates with a predominance of NCD (Turk-Kubo et al., 2014, Martinez-Perez et al., 2016; Moreiro-Coello et al., 2017, Fonseca-Batista et al., 2019; Tang et al., 2019).

### 4.3 Intriguing station 10

The patchy distribution of the diazotrophic activity at station 10 was related to an exceptionally high rate at the DCM (72.1 nmol N L$^{-1}$ d$^{-1}$). High rates compared to the low rates (< 0.5 nmol N L$^{-1}$ d$^{-1}$ usually over the whole water column measured in the MS) have previously been observed locally: 2.4 nmol N L$^{-1}$ d$^{-1}$ at the Strait of Gibraltar (Rahav et al., 2013a), ~5 nmol N





L$^{-1}$ d$^{-1}$ in the Bay of Calvi (Rees et al., 2017), 17 nmol N L$^{-1}$ d$^{-1}$ in the northwestern MS (Garcia et al., 2006) and 129 nmol N L$^{-1}$ d$^{-1}$ in the eastern MS (Rees et al., 2006). Station 10 was also hydrodynamically "contrasted" compared to the other

stations: it was located almost at the centre of an anticyclonic eddy (Guieu et al., 2020), with the core waters (0-200 m) of Atlantic origin (colder, fresher). In such anticyclonic structures, enhanced exchanges with nutrients rich waters from below take place, and combined with lateral mixing, could explain the high stocks of NO$_3^-$ and DIP in the euphotic layer (Table S1). Nevertheless, the anomaly of N$_2$ fixation at the DCM was neither associated with anomalies of PP, BP nor NO$_3^-$ and DIP concentrations. It only coincided with a minimum in DFe concentration (0.47 nM compared to 0.7 to 1.4 nM at the nearby

depths, Bressac et al., 2021) which could not be explained solely by the diazotrophs uptake.

Despite no correlation between N$_2$ fixation and the relative abundance of specific diazotrophs (p > 0.05) along the profile, the huge heterogeneity in N$_2$ fixation rate was likely related to the patchy distribution of diazotrophs taxa. Indeed, patchiness seems to be a common feature of unicellular diazotrophs (Robinart et al., 2014; Moreira-Coello et al., 2019). The exceptionally high N$_2$ fixation rates coincided with the highest relative contributions of UCYN-A and more precisely

UCYN-A4. Exceptional N$_2$ fixation rates at station 10, impacted by northeast Atlantic surface waters of subtropical origin could thus be related to that incoming waters. Indeed, Fonseca-Batista et al. (2019) reported high N$_2$ fixation rates (45 and 65 nmol N L$^{-1}$ d$^{-1}$ with euphotic N$_2$ fixation rates up to 1533 µmolN m$^{-2}$ d$^{-1}$) associated with a predominance of UCYN-A in subtropical Atlantic surface water mass along the Iberian Margin (~40° N-11° E). It should be noted that UCYN-A4 was only detected at station 10, and its relatively high contribution to the whole diazotrophs in the euphotic layer coincided with

the highest stocks of P (and N). This could reflect higher nutrient requirement(s) of the UCYN-A4 and/or of its eukaryotic partner relative to other sublineages. Another intriguing feature was the high contribution (~86 %) of UCYN-A in the mesopelagic zone (200 m). As UCYN-A lives in obligate symbiosis with haptophytes from which it receives fixed carbon from photosynthesis (Thompson et al., 2012, 2014), this suggests that this contribution was probably derived from sinking senescing prymnesiophyte-UCYN-A cells, and that the weak N$_2$ fixation rate at 200m depth is likely only driven by γ-

proteobacteria (*Pseudomonas*).

**4.4 Supply of bioavailable N from diazotrophic activity for fueling primary and heterotrophic bacterial production - Relationship with potential controlling factors of N$_2$ fixation**

The relationship established between N$_2$ fixation, and PP and BP illustrated that in the studied area, N$_2$ fixation is promoted

by UCYN and NCD, and/or could indicate that all processes have the same (co)-limitation. Overall, N$_2$ fixation was a poor contributor to PP, as previously shown in the MS (Bonnet et al., 2011; Yogev et al., 2011; Rahav et al. 2013a) and BP (1.0 ± 0.3 % of PP and 8 ± 1 % of BP) except at station 10 where N$_2$ fixation could support up to 19 % of PP and supply the entire bioavailable N requirements for heterotrophic prokaryotes (213 % of BP). As expected, our results suggest no control of N$_2$ fixation by DFe and NO$_3^-$, as previously shown through nutrient additions in microcoms (Rees et al., 2006; Ridame et al.,

2011, Rahav et al., 2016b). No correlation was observed between N$_2$ fixation and DIP which may highlight the spatial variability of the controlling factor of diazotrophs as DIP was shown to control N$_2$ fixation in the western basin, but not in





the Ionian basin (Ridame et al., 2011). Moreover, DIP concentration does not reflect the rapid turnover of P in the open MS and thus could be a poor indicator of DIP availability (Pulido-Villena et al., 2021).

**4.5 Diazotrophic activity and composition in response to dust addition under present climate conditions**

*General features* – In all experiments, simulated wet dust deposition under present climate conditions triggered a significant (41 to 503 %) and rapid (24-48 h) stimulation of $N_2$ fixation relative to the controls. Despite this strong increase, $N_2$ fixation rates remained low ($< 0.7$ nmol N $L^{-1}$ $d^{-1}$) as well as their contribution to PP ($< 7$ %) and BP ($< 5$ %) as observed *in situ* (Sect.4.4). All of these results are consistent with those found after dust seeding in mesocosms in a coastal site in the

northwestern MS (Ridame et al., 2013) and in the open Cretan Sea (Rahav et al., 2016a).

*Temporal Changes in the composition of the diazotrophic community-* Dust addition under present climate conditions did not impact the diazotrophic communities composition. At TYR and ION, the large increase in $N_2$ fixation recorded after dust addition might be attributed to NCD (mainly γ-proteobacteria), as suggested by the positive correlation between $N_2$ fixation and BP. At FAST, the community shifted from a large dominance of UCYN-A towards a dominance of NCD both in the

dust treatments and unamended controls due to the increase in a few fast growing γ-proteobacteria (mainly *Pseudomonas*). This shift could be attributed to a bottle effect imposed by the tanks which can favor fast growing heterotrophic bacteria (Sherr et al. 1999; Calvo-Diaz et al., 2011). Nevertheless, the increased $N_2$ fixation after dust seeding at FAST cannot be explained by the shift in composition of the diazotrophic communities because the rates remained quite stable in the controls all along the experiment. Rather, the abundances of diazotrophs have obviously increased due to dust input, and UCYN-A in

association with prymnesiophytes could still be responsible for the majority of the enhanced $N_2$ fixation as $N_2$ fixation correlated strongly with PP.

*Variability of the $N_2$ fixation response among stations* - The highest stimulation of $N_2$ fixation to dust addition was observed at TYR (mean $RC_D = 321$ %) then at ION (mean $RC_D = 161$ %) and finally at FAST (mean $RC_D = 21$ %) (Fig.7). The differences in the intensity of the diazotrophic response were not related to differences in the initial nutrients stocks

(Table S1) and in the nutrients input from dust which was quite similar between experiments (Gazeau et al., 2021a). Briefly, dust input led to a strong increase of $11.2\pm0.2$ μM $NO_3^-$ few hours after seeding in the three experiments, and the maximum DIP release was slightly higher at FAST (31 nM) than at TYR and ION ($23 \pm 2$ nM) (Gazeau et al., 2021a). As DFe concentration before seeding was high ($\geq 1.5$ nM, Table 2), the bioavailability of Fe did not appear to drive the response of $N_2$ fixation (Ridame et al., 2013). Also, we evidenced in this experiment that $NO_3^-$ release from dust did not inhibit $N_2$

fixation rate driven by UCYN-A and NCD. This was expected for UCYN-A as it lacks $NO_3^-$ assimilation pathways (Tripp et al., 2010; Bombar et al., 2014).

$N_2$ fixation was initially more limited at TYR and ION (as evidenced by the lowest initial rates) compared to FAST, thereby explaining the highest stimulation of $N_2$ fixation to dust seeding at these stations. Interestingly, the stimulation of $N_2$ fixation was higher at TYR than at ION (Fig.7) while these stations presented the same initial rate supported by NCD. One major

difference is that PP was not enhanced by dust seeding at TYR while BP increased in both experiments (Gazeau et al.,



2021b) suggesting that NCD-supported $N_2$ fixation was not limited by organic carbon at this station. As $N_2$ fixation and BP correlated strongly after the dust seeding, it means that dust-derived DIP could relieve the ambient limitation of both heterotrophic prokaryotes (BP was co-limited by NP, Van Wambeke et al., 2021) and NCD at TYR. This could explain why DIP concentration in the D treatments became again similar to the controls at the end of this experiment. At ION

characterized by the lowest initial DIP concentration, $N_2$ fixation and PP were likely DIP (co-)limited as shown for BP (Van Wambeke et al., 2021). Consequently, heterotrophic prokaryotes, NCD, and photoautotrophs could out-compete for dust-derived DIP uptake reducing then the amount of DIP per cell explaining the lowest stimulation of $N_2$ fixation relative to TYR.

At FAST, initially dominated by UCYN-A, $N_2$ fixation and PP correlated strongly after the dust seeding. This indicated that

dust could relieve either directly the ambient nutrient limitation of both $N_2$ fixation and PP (Fig.S3) or indirectly through first the relief of the PP limitation of the UCYN-A photoautotroph hosts inducing an increase in the production of organic carbon which could be used by UCYN-A to increase its $N_2$-fixing activity.

### 4.6 Response to dust addition under future relative to present climate conditions

*General features* -At TYR and FAST, $N_2$ fixation was more stimulated by dust input under future than present climate conditions (mean $RC_{G-TYR}$= 478 % and mean $RC_{G-FAST}$= 54 %) whereas at ION the response was similar (Figs.7, S2). These differences between future and present climate conditions were not related to the nutrients supplied from dust (Gazeau et al., 2021a).

The purpose of our study was to study the combined effect of warming and acidification, but we can expect on the short time

scale of our experiments (< 3-4 days), that NCD and UCYN-A would not be directly affected by ocean acidification and the associated changes in the $CO_2$ concentration as they do not fix $CO_2$ (Zehr et al., 2008). Indeed, no impact of acidification (or $pCO_2$ increase) on $N_2$ fixation was detected when the diazotrophic communities were dominated by UCYN-A in the North and South Pacific (Law et al., 2012; Böttjer et al. 2014).

*TYR and ION* –Under future climate conditions, the composition of the diazotrophic communities did not change after dust input at TYR and ION relative to present conditions. At TYR, the highest $N_2$ fixation stimulation might be linked to the increase in the NCD abundances and/or in their cell-specific $N_2$ fixation rates under future climate conditions. Unfortunately, the impact of increased temperature and decreased pH on the cell-specific $N_2$ fixation rates of NCD is currently unknown. However, some studies suggest a positive relationship between temperature and abundances of NCD: diazotrophic γ-

proteobacteria (γ-24774A11) gene copies correlated positively with temperature (from ~20 to 30° C) in surface waters of the western South Pacific Ocean (Moisander et al., 2014), and Messer et al. (2015) suggested a temperature optima for these γ-proteobacteria around 25-26° C in the Australian tropical waters. At ION, the similar stimulation of $N_2$ fixation by dust under future climate conditions compared to present climate conditions could be explained by a greater mortality of diazotrophs due to a higher grazing pressure and/or a higher viral activity. Indeed, higher bacterial mortality in the G





treatment that could be related to a higher grazing pressure has been observed (Dinasquet et al., 2021). Another explanation is that in spite of the DIP supply from the dust, the DIP bioavailability, initially the lowest at ION, was not sufficient to allow an additional $N_2$ fixation stimulation.

*FAST-* Some differences in the composition of the diazotrophic communities were observed between present and future climate conditions at FAST after dust input: the contribution of NCD (mainly *Pseudomonas)* increased and that of UCYN-A

decreased. It must be noted that the duration of the experiment was longer at FAST (4 days) relative to TYR and ION (3 days) which could explain at least partly differences between stations. A direct response of increased temperature and/or decreased pH can be considered on a very short time scale (12 hours) by comparing the results in the G treatment at T0 (+3° C, -0.3 pH unit) with those in C and D treatments. The increased contribution of *Pseudomonas* in the G treatment at T0 (before dust addition) reveals a likely positive effect of temperature on the growth of this NCD instead of a decrease in the

top-down control on the bacterioplankton which is strongly suspected to increase under future climate conditions (Dinasquet et al., 2021). Interestingly, despite the decrease in the contribution of UCYN-A after dust addition, we observed contrasted responses within the UCYN-A pool relative to present climate conditions: the UCYN-A3 contribution strongly decreased (4.6 % in G vs. 25.4 % in D) whereas that of UCYN-A2 was twice as high (7 % in G vs. 3.4 % in D) (UCYN-A1 contribution was similar). These respective changes could be explained by the difference in the temperature tolerance

between UCYN-A2 and -A3. Temperature is one of the key drivers explaining the distribution of UCYN-A which appeared to dominate in most of the temperate regions with temperature optima around ~22-24° C (Langlois et al., 2008; Moisander et al., 2010). However, the temperature optima for the different UCYN-A sublineages, in particular for UCYN-A2 and -A3, are poorly known. Interestingly, Henke et al. (2018) observed that the UCYN-A2 abundance was positively affected by increasing temperature, within a range of temperature from about 21 to 28° C which is in agreement with our results. Based

on the strong positive correlation between $N_2$ fixation and PP after dust addition, and despite the decrease in the relative abundance of UCYN-A3, the increased stimulation of $N_2$ fixation under future climate conditions could be sustained by the increase in the relative abundance of UCYN-A2 which is bigger than UCYN-A3 (Cornejo-Castillo et al., 2019) and could consequently have a higher cell-specific $N_2$ fixation rate.

**5. Conclusion**

In the MS, $N_2$ fixation is a minor pathway to supply new bioavailable N for sustaining both PP and BP but can locally support up to 20 % of PP and provide all the N requirement for bacterial activity. UCYN-A could support extremely high rates of $N_2$ fixation (72 nmol.L$^{-1}$.d$^{-1}$) in the core of an eddy in the Algerian basin influenced by Atlantic waters. The eastward decreasing longitudinal trend of $N_2$ fixation in the surface waters is likely related to the spatial variability of the composition

of the diazotrophic communities, as shown by the eastward increase in the relative abundance of NCD towards more oligotrophic waters while we observed a westward increase in the relative abundance of UCYN-A. This could reflect lower nutrients requirements for NCD relative to UCYN-A. Through the release of new nutrients, simulated wet dust deposition under present and future climate conditions significantly triggered $N_2$ fixation. The degree of stimulation depended on the
metabolic activity of the diazotrophs (degree of limitation) related to the composition of diazotrophic communities, and on
the ambient potential nutrient limitations of diazotrophs, including that of the UCYN-A prymnesiophyte host. The strongest
increase in $N_2$ fixation, not accompanied with a change in the composition of the diazotrophic communities, was observed at
the stations dominated by NCD (TYR, ION) where the nutrient limitation was the strongest. Under projected future levels of
temperature and pH, the dust effect is either exacerbated or unchanged. Knowing that NCD and UCYN-A do not fix $CO_2$, we
suggest that, on the time scale of our experiments (3-4 days), the exacerbated response of $N_2$ fixation is likely the result of
the warming (from about 21° C to 24° C) which may increase the growth of NCD when nutrient availability allows it, and
may alter the composition of UCYN-A community. However, to date, the effect of acidification and temperature optima of
the different UCYN-A sublineages are poorly known (or unknown) as these UCYN-A remained uncultivated.

Future changes in climate, desertification and land use practices could induce an increase in dust deposition to the oceans
(Tegen et al., 2004; Moulin and Chiapello, 2006; Klingmüller et al., 2016). The predicted future increase in surface
temperature, and the resulting stronger stratification would expect to expand the surface of LNLC areas reinforcing
consequently the role of new nutrient supply from aeolian dust on the $N_2$ fixation and probably on the structure of the
diazotrophic communities.

## 6. Data availability

Guieu,  C.  et  al.  (2020).  Biogeochemical  dataset  collected  during  the  PEACETIME  cruise.  SEANOE.
https://doi.org/10.17882/75747.

## 7. Author contributions

FG and CG designed the dust seedings experiments. CR, JD, EB, MB, FVW, FG, VT, AT-S and CG participated to the
sampling and analysis. CR and EB performed DNA extraction; EB performed library preparation. CR, JD and SH analyzed
the data; CR wrote the manuscript with contributions from all authors.

## 8. Competing interests

The authors declare that they have no conflict of interest


## 9. Special issue Statement

This article is part of the special issue "Atmospheric deposition in the low-nutrient–low-chlorophyll (LNLC) ocean: effects
on marine life today and in the future (ACP/BG inter-journal SI)". It is not associated with a conference.

## 10. Financial support

This study is a contribution to the PEACETIME project (http://peacetime-project.org), a joint initiative of the MERMEX and
ChArMEx  components  supported  by  CNRS-INSU,  IFREMER,  CEA,  and  Météo-France  as  part  of  the  programme



MISTRALS coordinated by INSU. PEACETIME was endorsed as a process study by GEOTRACES. JD was funded by a Marie Curie Actions-International Outgoing Fellowship (PIOF-GA-2013-629378). SH and LR were funded by grant 6108-00013 from the Danish Council for independent research to LR.

## 11. Acknowledgments

The authors thank the captain and the crew of the RV Pourquoi Pas ? for their professionalism and their work at sea. We warmly acknowledge our second 'chieffe' scientist Karine Desboeufs. We gratefully thank Eric Thiebaut and Pierre Kostyrka for their precious advice with statistical tests. We also thank Kahina Djaoudi and Thibaut Wagener for their assistance in sampling the tanks and TMC-rosette.

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





**Table 1:** Integrated $N_2$ fixation over the surface mixed layer (SML, from surface to the mixed layer depth), from the surface
to the base of the euphotic layer (1% PAR depth), over the aphotic layer (1%PAR depth to 1000 m), and from surface to
1000 m at all the sampled stations. Contribution (in %) of SML integrated $N_2$ fixation to euphotic layer integrated $N_2$
fixation, and contribution of euphotic layer integrated $N_2$ fixation to total (0-1000 m) integrated $N_2$ fixation.

| | $N_2Fix_{SML}$ | $N_2Fix_{euphotic}$ | $N_2Fix_{aphotic}$ | $N_2Fix_{0-1000m}$ | $N_2Fix_{SML}/N_2Fix_{euphotic}$ | $N_2Fix_{euphotic}/N_2Fix_{0-1000m}$ |
|---|---|---|---|---|---|---|
| | µmolN m$^{-2}$ d$^{-1}$ | µmolN m$^{-2}$ d$^{-1}$ | µmolN m$^{-2}$ d$^{-1}$ | µmolN m$^{-2}$ d$^{-1}$ | % | % |
| ST01 | 14.6 | 42.6 | 56.5 | 99.1 | 34 | 43 |
| ST02 | 10.7 | 36.0 | 16.0 | 51.9 | 30 | 69 |
| ST03 | 7.8 | 58.3 | 18.1 | 76.4 | 13 | 76 |
| ST04 | 10.8 | 46.6 | 38.5 | 85.1 | 23 | 55 |
| ST05 | 4.9 | 46.3 | 36.1 | 82.4 | 10 | 56 |
| TYR | 4.2 | 38.6 | 53.0 | 91.6 | 11 | 42 |
| ST06 | 9.1 | 34.9 | 29.8 | 64.7 | 26 | 54 |
| ST07 | 10.5 | 43.5 | 55.4 | 98.8 | 24 | 44 |
| ION | 6.2 | 40.6 | 56.5 | 97.1 | 15 | 42 |
| ST08 | 4.3 | 27.0 | 12.3 | 39.3 | 16 | 69 |
| ST09 | 3.4 | 50.2 | 43.3 | 93.5 | 7 | 54 |
| FAST | 5.9 | 58.2 | 35.7 | 93.8 | 10 | 62 |
| ST10 | 13.7 | 1908 | 63.7 | 1972 | 1 | 97 |
| Mean ± std (ST10 excluded) | 7.7±3.5 | 44±9 | 38±16 | 81±20 | | |
| Mean ± std (all stations) | | | | | 17%±10% | 59%±16% |







**Table 2**: Initial physico-chemical and biological properties of surface seawater before the perturbation in the dust seeding experiments at TYR, ION and FAST (average at T0 in C and D treatments, n=4 or data at T-12h in the pumped surface waters, n=1). The relative abundances of diazotrophic cyanobacteria and NCD (non-cyanobacterial diazotroph) are given as proportion of total *nifH* sequence reads. DIP: dissolved inorganic phosphorus, DFe: dissolved iron. Means that did not differ significantly between the experiments (p>0.05) are labeled with the same letter (in parenthesis).

| | TYR | ION | FAST |
|---|---|---|---|
| Day of sampling | 05/17/2017 | 05/25/2017 | 06/02/2017 |
| Temperature (° C)* | 20.6 | 21.2 | 21.5 |
| Salinity* | 37.96 | 39.02 | 37.07 |
| $^{13}$C-Primary production, mg C m$^{-3}$ d$^{-1}$ | 1.23±0.64 (A) | 2.53±0.40 (B) | 2.82±0.55 (B) |
| $N_2$ fixation nmol N L$^{-1}$ d$^{-1}$ | 0.19±0.03 (A) | 0.21±0.05 (A) | 0.51±0.04 (B) |
| Relative abundance of diazotrophic cyanobacteria (%) | 4.7±3.8 (A) | 6.2±6.5 (A) | 91.4±6.0 (B) |
| Relative abundance of NCD (%) | 95.3±3.9 (A) | 93.8±6.5 (A) | 8.6±6.0 (B) |
| Heterotrophic bacterial production ng C. L$^{-1}$ h$^{-1}$ | 26.6 ± 7.0 (AB) | 25.9 ± 0.9 (A) | 36.3 ± 1.2 (B) |
| C:N (mol/mol) | 9.6±0.8 (A) | 10.2±0.8 (A) | 9.1±0.5 (A) |
| DIP, nM* | 17 | 7 | 13 |
| $NO_3^-$, nM* | 14 | 18 | 59 |
| $NO_3^-$/DIP, mol/mol | 0.8 | 2.6 | 4.5 |
| DFe, nM$^§$ | 1.5±0.1 (A) | 2.6±0.2 (B) | 1.8±0.2 (A) |

\* from Gazeau et al., 2021a

§ from Roy-Barman et al., 2021







**Figures captions**

Figure 1: Locations of the ten short (ST1 to ST10) and three long stations (TYRR, ION and FAST). Stations 1 and 2 were located in the Provencal basin; stations 5, 6, and TYR, in the Tyrrhenian Sea; stations 7, 8, and ION in the Ionian Sea; and stations 3, 4, 9, 10 and FAST in the Algerian basin. Satellite-derived chlorophyll-a concentration (mg m$^{-3}$) averaged over the entire duration of the PEACETIME cruise. Ocean color data from MODIS/Aqua, NASA.

Figure 2: Vertical distribution of N$_2$ fixation (in nmol N L$^{-1}$ d$^{-1}$) in the Provencal (a), Tyrrhenian (b) Ionian (c) and Algerian (d) basins and at Station 10 (e). N$_2$ fixation rates at station 10 are plotted individually because of the high fluxes.

Figure 3: Surface (a) and integrated N$_2$ fixation from surface to euphotic layer depth (b) along the longitudinal PEACETIME transect (station 10 was excluded).


Figure 4: Vertical distribution of the 20 most abundant nifH-ASVs at station 10, collapsed into major taxonomic groups.

Figure 5: N$_2$ fixation rate integrated over the euphotic layer versus $^{13}$C-primary production (a) and bacterial production (b); data at station 10 were removed.

Figure 6: N$_2$ fixation rate in nmol N L$^{-1}$ d$^{-1}$ during the dust seeding experiments performed at the stations TYR (a), ION (b) and FAST (c) in the replicated controls (black dot), dust treatments under present climate conditions (red square, D treatment) and dust treatments under future climate conditions (green triangle, G treatment). Open symbols were not included in the linear regression

Figure 7: Box plots of the relative changes (in %) in N$_2$ fixation to the rates measured in the controls over the duration of the dust seeding experiments at TYR, ION, and FAST stations. D means dust treatments under present climate conditions (D treatment) and G dust treatments under future climate conditions (G treatment). The red cross represents the average.

Figure 8: The composition of diazotrophs (based on 20 most abundant ASVs in the tanks) during the dust seeding 890 experiments at the start (T0) and end (T3 at TYR and ION, and T4 at FAST) in each tank, at TYR (Top panel), ION (middle panel) and FAST (bottom panel). C1T0 at TYR was not included due to poor sequencing quality.

Figure 9: Relative abundance of the 20 most abundant nifH-ASVs in surface waters (values at TYR, ION and FAST are based on average of duplicated control and dust treatments at T0).







**Figure 1**








**Figure 2**

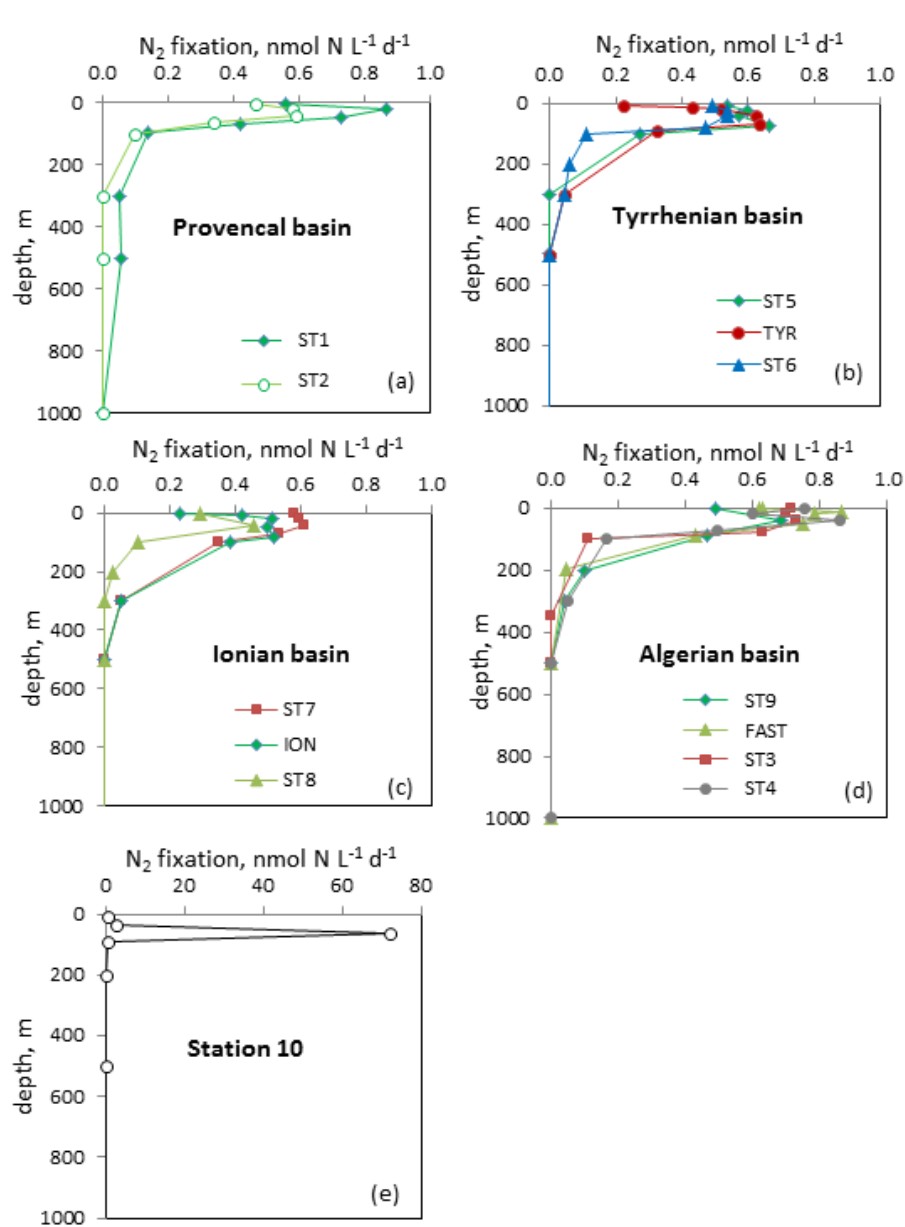



**Figure 3**

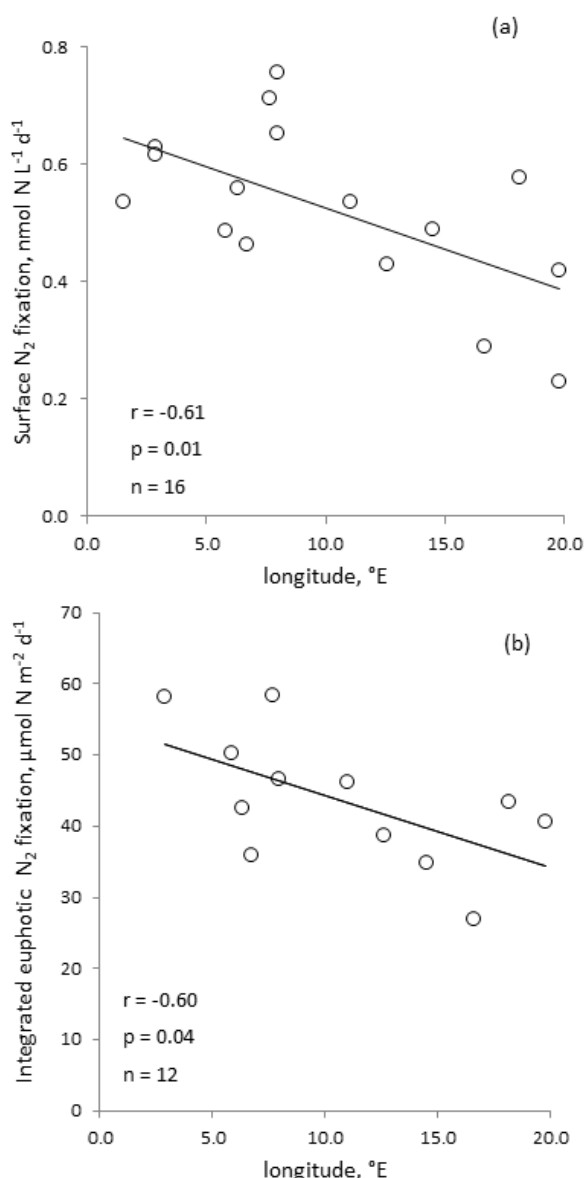





**Figure 4**


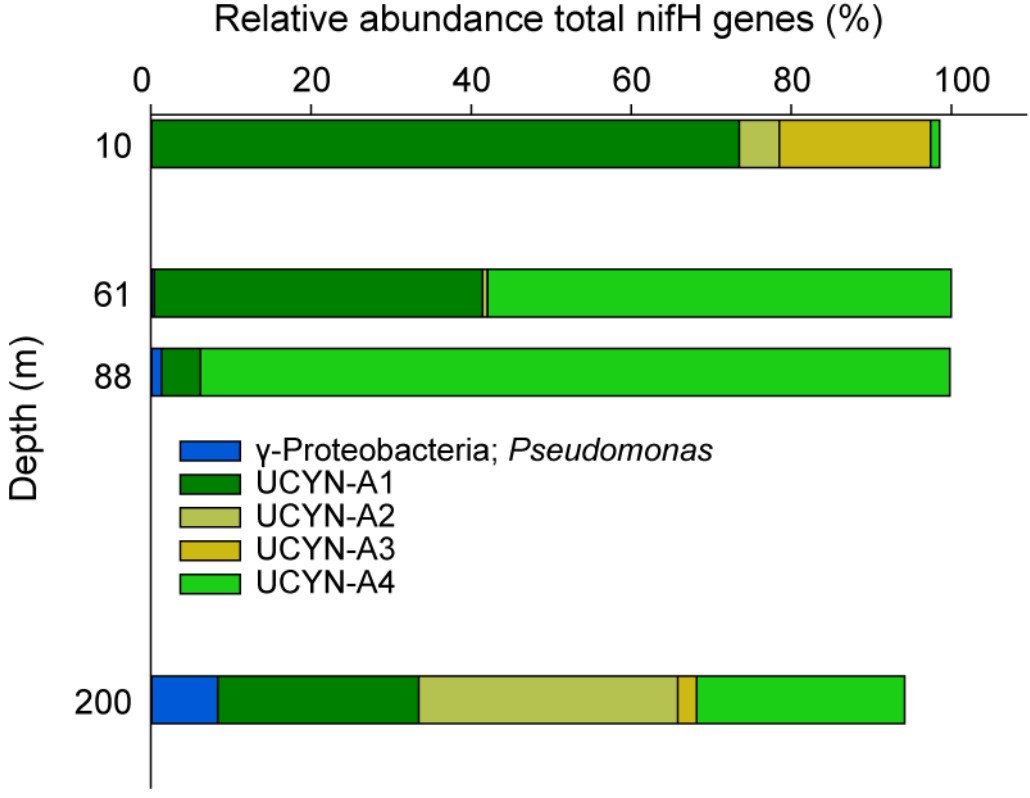








**Figure 5**

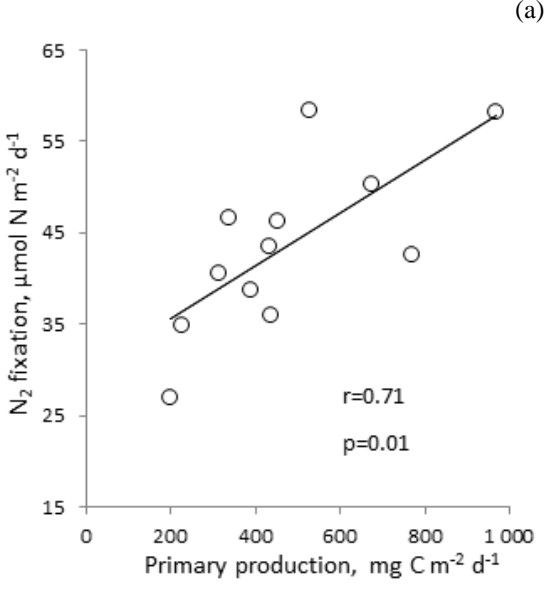

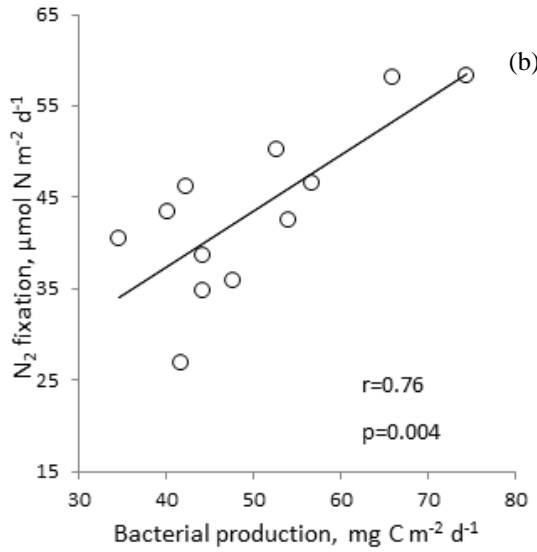






**Figure 6**

(a)

(b)

(c)





**Figure 7**

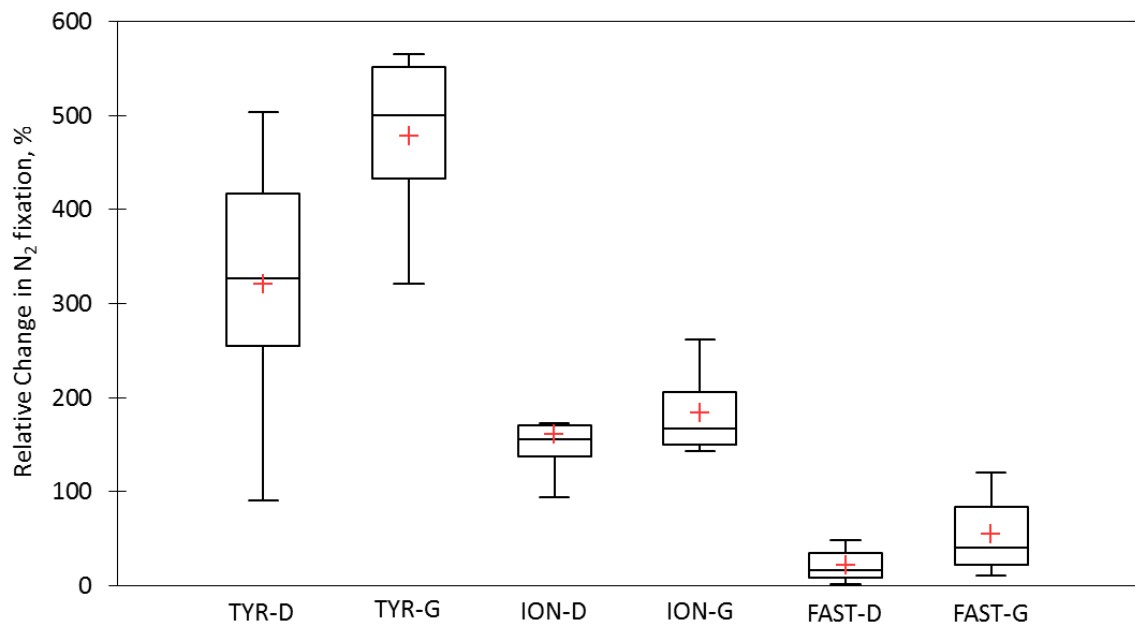








**Figure 8**







**Figure 9**


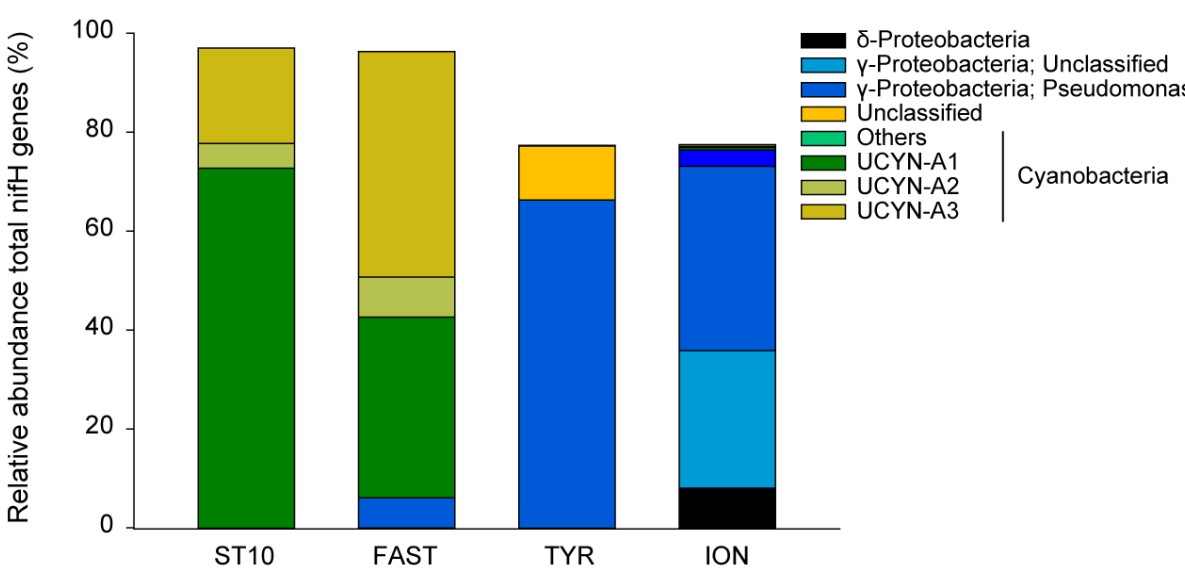




