# Peer review of "N2 fixation in the Mediterranean Sea related to the composition of the diazotrophic community, and impact of dust under present and future environmental conditions"

_Biogeosciences, 2021_

## Author Comment (AC1)

We appreciate the reviewers' interest and thank them for their relevant suggestions and time spent reviewing this manuscript

.**REPLY to referee RC1**

 **GENERAL COMMENTS:**

- **Q1** It is now generally accepted that minimum detectable uptake rates ($N_2$ and $CO_2$) should be determined for every individual incubation experiment, so that rates under their specific detection limit can be reported as such (<DL). Because every sampling site and sampling depth (and sampling time) have their own original substrates concentrations and associated isotope compositions (PN, POC, dissolved $N_2$ and dissolved inorganic carbon), it makes it important to compute incubation-specific minimum detectable uptake rate, based on the minimum increase in isotope composition detectable by the isotope ratio mass spectrometer. The authors should confirm that all reported $N_2$ fixation rates are indeed truthful (particularly at depths $\geq$ 200 m).
  **Lines 149-150:** The authors should confirm that all reported $N_2$ fixation rates are indeed truthful (particularly at depths $\geq$ 200 m), by computing incubation-specific minimum detectable uptake rates, based on the minimum increase in isotope composition (relative to natural abundance), detectable by the isotope ratio mass spectrometer (Fonseca-Batista et al., 2017; White et al., 2020).

Reply to RC1: We agree with the reviewer that some details regarding N2 fixations rates were missing. The following additions (in bold) are now included in the text in MM, section 2.3:

**'After collection, 2.3 L of seawater were immediately filtered onto pre-combusted GFF filters to determine natural concentrations and isotopic signatures of particulate organic carbon (POC) and particulate nitrogen (PN).** Net $N_2$ fixation rates were determined using the $^{15}N_2$ gas-tracer addition method (Montoya et al., 1996), and net primary production using the $^{13}C$-tracer addition method (Hama et al., 1983). Immediately after sampling, 1 mL of $NaH^{13}CO_3$ (99 %, Eurisotop) and 2.5 ml of 99 % $^{15}N_2$ (Eurisotop) were introduced to 2.3 L polycarbonate bottles through a butyl septum for simultaneous determination of $N_2$- and $CO_2$-fixation. $^{15}N_2$ and $^{13}C$ tracers were added to obtain a ~10 % final enrichment.'.....'After 24 h incubation, 2.3 L were filtered onto pre-combusted 25 mm GF/F filters, and filters were stored at −25° C. Filters were then dried at 40° C for 48 h before analysis. POC and PN as well as $^{15}N$ and $^{13}C$ isotopic ratios were quantified using an online continuous flow elemental analyzer (Flash 2000 HT), coupled with an Isotopic Ratio Mass Spectrometer (Delta V Advantage via a conflow IV interface from Thermo Fischer Scientific). **For each sample, POC (in the 0-100m layer) and PN (0-1000m) were higher than the analytically determined detection limit of 0.15 µmol for C and 0.11 µmol for N. Standard deviations were 0.0007 atom% and 0.0005 atom% for 13C and 15N enrichment, respectively. The atom% excess of the dissolved inorganic carbon (DIC) was calculated by using measured DIC concentrations at the LOCEAN laboratory (SNAPO-CO2).** N2 fixation rates were calculated by isotope mass balance equations as described by Montoya et al. (1996). **For each sample, the 13C and 15N uptake rates were considered as significant when excess enrichment of POC and PN was greater than three times the standard deviation obtained on natural samples. According to our experimental conditions, the minimum detectable 13C and 15N**

**uptake rates in our samples were 5 nmol C L$^{-1}$ d$^{-1}$ and 0.04 nmol N L-1 d-1 respectively. CO$_2$ uptake rates were above the detection limit in the upper 0-100m, while N2 fixation was not quantifiable below 300 m depth except at stations 1 and 10 with rates ~0.05 nmol N L$^{-1}$ d$^{-1}$at 500 m depth.'** (see in the submitted version, Results L225-227)

For the sake of clarity, we have symbolized by crosses the N2 fixation rates under detection limit ($<0.04$ nmol N L$^{-1}$ d$^{-1}$) on Fig. 2.

- **Q2**

The authors should be more skeptical and critical when comparing high-throughput sequencing of *nifH* gene from different sampling depth, sites and time points- The authors should be more skeptical when comparing relative abundance data from different sampling depth, sites and time points (Gloor et al., 2017): Lines 290-292, 447-450 and 462-463.

Reply to RC1: We appreciate this insightful comment. The authors are well aware of the issues and potential caveates inherent in the analysis of compositional microbiome datasets. However, we do see that the choice of words in the lines mentioned could have been selected more carefully to avoid any speculation of overinterpretation. The following additions (in bold) are now included in the text

Lines 290-292, section 3.2.3 At FAST, no difference in the relative abundances of diazotrophs was recorded between D treatment and the controls at T4. **However, when comparing G treatment relative to D at T4, the relative contribution of NCD was higher (82 % in G vs. 63 % in D) and the relative abundance of UCYN-A was lower** (13 % in G vs. 31 % in D).

447-450 discussion 4.6 'Interestingly, despite the decrease in the **relative** contribution of UCYN-A **to the total diazotrophs community** after dust addition, we observed contrasted responses within the UCYN-A group relative to present climate conditions: **the relative abundance of** UCYN-A3 strongly decreased (4.6 % in G vs. 25.4 % in D) whereas **the relative abundance of UCYN-A2 was twice as high (7 % in G vs. 3.4 % in D). Notably, the relative contribution of UCYN-A1 did not appear to be impacted during the dust addition experiment.**

462-463 conclusions 'UCYN-A **might be supporting** extremely high rates of N$_2$ fixation (72 nmol.L$^{-1}$.d$^{-1}$) in the core of an eddy in the Algerian basin influenced by Atlantic waters.

- **Q3**

Primary production rates measurements (based on the $^{13}$C incubation method), although mentioned all along the manuscript (with relation to corresponding N$_2$ fixation rates) are not described or discussed. The authors invite the reader to report to the manuscript by Maranon et al. (2020), who used a different methodology ($^{14}$C incubation technique). The authors should inform the reader (e.g., in the supporting information) about how the results from the two methods compare? Whether they show similar trends across the sampling sites and dust seeding experiments, despite the contrast in gross versus net rate assessments? This would support the authors' decision not to further discuss primary

production in their manuscript and invite the reader to report to Maranon et al. (2020) for more detailed insights. **M&M-Lines 139-140:** The authors should indicate in the supplementary information how consistent the results from the two methods are ($^{13}$C-PP and $^{14}$C-PP). As of now, no other manuscript in the Special Issue describes or discusses the $^{13}$C-PP rate measurements. Unless a manuscript comparing the data from the two methods is envisioned, having a brief comparison in the Supplementary Material would support the authors' choice not to discuss $^{13}$C-PP further in this manuscript and focus on $N_2$ fixation and diazotrophic community compositions.

Reply to RC1: For the sake of clarity, we have added this paragraph and this figure in Supplementary Information

'Figure S1: Comparison between $^{13}$C-PP and $^{14}$C-PP measured in the particulate matter during the dust seeding experiments

In situ samples for $^{13}$C-PP and $^{14}$C-PP were not systematically measured at the same depths (±10 m) and on the same day; seawater for $^{14}$C-PP was collected with the classical rosette (Niskin bottles) (Maranon et al., 2021) while $^{13}$C-PP seawater was sampled with the trace metal clean rosette. We therefore chose to use $^{13}$C-PP data to estimate the contribution of $N_2$ fixation to PP because both parameters were measured simultaneously in the same bottle. Nevertheless the shapes of the profiles and trends are similar with both data sets. In addition, $^{14}$C-PP (Gazeau et al., 2021b) and $^{13}$C-PP were measured in parallel during the dust seeding experiments and the correlation between $^{13}$C-PP and $^{14}$C-PP values was very strong (r=0.97, p<0.0001, n=72) as shown in the figure below'

[Figure]

We have also added in the revised version in MM (in bold): '*In situ* 13C-PP will not be discussed in this paper as 14C-PP rates are presented in Maranon et al. (2021) (**see details in Fig. S1**). The *in situ* 13C-PP were used in the present study to estimate the contribution of N2 fixation to PP **as both parameters were measured simultaneously**'

- **Lines 139-140 and 152-153:** Please clarify for the reader that the contribution of $N_2$ fixation to primary production and to bacterial production where estimated using C:N Redfield ratio (6.6) and ratio from Nagata (1986), respectively.

Reply to RC1: We didn't use the Redfield ratio (6.6) to estimate the contribution of $N_2$ fixation to PP. Instead, we used the molar C/N ratio measured in the organic particulate matter of our samples by EA-IRMS (L146-147) as on each GFF sample, we measured 4 parameters : particulate carbon and nitrogen, and 13C and 15N isotopic ratios (as mentioned L151).

We have added in the revised version in MM (in bold) 'The *in situ* 13C-PP **and molar C/N ratio in the organic particulate matter in our samples (see below for details) measured simultaneously in our samples** were used to estimate the contribution of N2 fixation to PP **.'**

- **Line 152:** for the sake of clarity, please inform the reader that BP measurements, which methodology has at this stage not yet been described, are complementary data presented in companion manuscripts (Gazeau et al., 2021b; Van Wambeke et al., 2021) and **Lines 152-153:** have the authors considered citing Fukuda et al. (1998) (manuscript with Nagata Toshi himself as co-author), to support their choice of C:N conversion factor. In fact, the cell collection in Fukuda et al., seems more appropriate for bacteria than the GF/F filtration used in Nagata (1986), thereby leading to a more reliable estimate of the bacterial C:N ratio in oceanic settings of $6.8 \pm 1.2$.

Reply to RC1: We agree with the reviewer; the choice of a C/N ratio of 6.8 measured in oceanic bacterial assemblages is more appropriate. We have therefore recalculated the contribution of N2 fixation to BP using a molar C/N ratio of 6.8, and modified the contribution (%) of N2 fixation to BP which decreases slightly, in section 4.4 of the discussion (the contribution of N2 Fixation to BP in section 4.5 remains unchanged (from 5.1% to 4.8% so ~5%). The general conclusions (N2 fixation is a poor contributor to BP) remain unchanged.

Changes in the revised version in MM (in bold) 'As a rough estimate of the potential impact of bioavailable N input from $N_2$ fixation on BP, **we used the BP rates presented in companion papers (Gazeau et al., 2021b; Van Wambeke et al., 2021), and converted them** in N demand using the molar ratio C/N of **6.8 (Fukuda et al., 1998).'**

Changes in section 4.4 (in bold) '**Overall, $N_2$ fixation was a poor contributor to PP (1.0 $\pm$ 0.3 %), as previously shown in the MS (Bonnet et al., 2011; Yogev et al., 2011; Rahav et al. 2013a) and BP (7 $\pm$ 1 %)** except at station 10 where $N_2$ fixation could support up to 19 % of PP and supply the entire bioavailable N requirements for heterotrophic prokaryotes (**199 % of BP**).'

- **Line 149:** The authors chose to use of the $^{15}N_2$ bubble addition method for their incubation experiments, which has been shown to underestimate in situ $N_2$ fixation activity due to incomplete tracer dissolution. The authors clearly stated that. However, to alleviate some of this uncertainty, the authors could consider in the future, sampling the incubation bottles at the end of the experiment (before filtration) to determine the final $^{15}N\%-N_2$ enrichment, which can then be used to compute $N_2$ fixation rates. Although these rates would likely still underestimate the true activity (due to dissolution kinetics taking place during the 24-hour incubation), they would however reduce the uncertainty and inform on the gap between $N_2$ fixation rates based on measured versus theoretically estimated $^{15}N-N_2$ enrichments.

Reply to RC1: We are in complete agreement with the reviewer. Such measurement of 15N atom% of the dissolved 15N2 prior to filtration at the end of the incubation period would indeed provide a more accurate N2 fixation rate. We have to develop the measurement of the isotopic ratio of 15N2 under dissolved form with our IRMS. We also chose to use the 15N2 bubble addition method because some studies have shown trace element contamination with the 15N2 enriched water method

- Line 176: please explain what influenced the decision to truncate the reads at 350 bp? (no need to report in the manuscript)

Reply to RC1: we truncated the read at 350bp because the quality decreased for longer reads and we wanted to keep high quality scores. The expected length of the nifH amplicon is 362, therefore the sequence information lost is minimal. From the authors experience this will not impact the taxonomic classification obtained with the employed sequence analysis pipeline.

**Results**

- **Line 273:** "CV%" not previously defined

Revised to ' The reproducibility between the replicated treatments was good at all stations (**mean coefficient of variation (CV%)** $< 14$ %).

- **Line 281:** please clarify, "low overall **relative** abundance".

Revised to 'Some of these ASVs had low overall **relative** abundance,'

- **Line 286:** Specify from which condition(s) (Control, Dust and Greenhouse) the average contributions of UCYN-A1 and A3 to the total diazotrophic community were determined from at T0.

Revised to '(relative abundance of UCYN-A1 and -A3 **in C and D treatments at T0, n=4** : $34 \pm 6$ % and $45 \pm 2$ % of the total diazotrophic composition, respectively)'

**Discussion:**

**Lines 320-321, 361-362, 378-379, 401-402, 404, 409 and 455:** data not shown, that could be added to the Supplementary Material, with relation to:

1) correlation between $N_2$ fixation rates and diazotrophic community composition (for instance, surface $N_2$ fixation versus UCYN-A and NCD) (Lines 320-321)

Reply to RC1: we have added Fig S8 showing the relationship between surface $N_2$ fixation and (a) UCYN-A and (b) NCD

2) contribution of N$_2$ fixation to PP and BP

L361-362, Reply to RC1: we have added a new figure in SI (Fig S9) showing the contribution of N$_2$ fixation to PP and BP at the studied stations

[Figure]

L378-379, L401-402, L409 and L455 Reply to RC1: we have added a new figure in SI (Fig S4) showing the relationship between N$_2$ fixation and (a) BP at TYR, (b) BP at ION, and (c) PP at FAST, during the dust seeding experiments

3) evolution of nutrient concentration in the dust seeding experiments: DIP concentration in Control and Dust experiments at station TYR; requiring citation of the corresponding companion paper (line 404).

L404, section 4.5, we have added (in bold) 'This could explain why DIP concentration in the D treatments became again similar to the controls at the end of this experiment **(Gazeau et al., 2021a)'**.

**Lines 331-332:** Sentence not clear, please rephrase.

Revised to: 'High **$N_2$ fixation** rates have previously been observed locally: 2.4 nmol N L$^{-1}$ d$^{-1}$ at the Strait of Gibraltar (Rahav et al., 2013a), ~5 nmol N L$^{-1}$ d$^{-1}$ in the Bay of Calvi (Rees et al., 2017), 17 nmol N L$^{-1}$ d$^{-1}$ in the northwestern MS (Garcia et al., 2006) and 129 nmol N L$^{-1}$ d$^{-1}$ in the eastern MS (Rees et al., 2006).

**Line 340:** Please explain further why the DFe minimum could not only be the result of uptake by diazotrophs

Reply to RC1: We estimated the theoretical Fe requirement to sustain a N2 fixation of 72 nmol N L$^{-1}$ d$^{-1}$ at 61m, station 10 using a range (min-max) of Fe/C (from 7 to 177 µmol:mol) and associated C/N for diazotrophs ($Trichodesmium$, UCYN) from literature (Berman-Frank et al., 2007; Tuit et al., 2004, Jiang et al., 2018). We found that to sustain this $N_2$ fixation rate, 0.004 nM to 0.08 nM of DFe are required. Consequently, the minimum in DFe concentration at 61m of 0.47 nM compared to 0.7 to 1.4 nM at the nearby depths, (Bressac et al., 2021) could not be explained solely by the diazotrophs uptake.

We have added in the revised version (in bold): 'It only coincided with a minimum in DFe concentration (0.47 nM compared to 0.7 to 1.4 'nM at the nearby depths, Bressac et al., 2021). **Based on a range of Fe:C (from 7 to 177 µmol:mol) and associated C:N ratios for diazotrophs ($Trichodesmium$, UCYN) from literature (Berman-Frank et al., 2007; Tuit et al., 2004; Jiang et al., 2018 ), we found that 0.004 nM to 0.08 nM of DFe are required to sustain this $N_2$ fixation rate. Consequently, the minimum in DFe concentration at 61m could not be explained solely by the diazotroph uptake.**

**Line 445:** "a decrease in the top-down control on the bacterioplankton which is strongly suspected to increase under future climate conditions" Please explain further why

We rephrased this sentence to '**The increased contribution of *Pseudomonas* in the G treatment at T0 (before dust addition) reveals a likely positive effect of temperature on the growth of this NCD as an increase in the top-down control on the bacterioplankton was observed after dust seeding under future climate conditions (Dinasquet et al., 2021).'

**Conclusion:**

Lines 462-463: Because cell specific $N_2$ fixation rates were not determined, this statement should be less affirmative.

Reply to RC1: Please, See our response to general comment in Q2.

**Tables and Figures:**

- Table 1: Why were some average and standard deviation values not included in the two bottom rows?

The two bottom rows are now filled in the revised version;

Table 1: Integrated $N_2$ fixation over the surface mixed layer (SML, from surface to the mixed layer depth), from the surface to the base of the euphotic layer (1% PAR depth), over the aphotic layer (1%PAR depth to 1000 m), and from surface to 1000 m at all the sampled stations. Contribution (in %) of SML integrated $N_2$ fixation to euphotic layer integrated $N_2$ fixation, and contribution of euphotic layer integrated $N_2$ fixation to total (0-1000 m) integrated $N_2$ fixation.

| | $N_2Fix_{SML}$ | $N_2Fix_{euphotic}$ | $N_2Fix_{aphotic}$ | $N_2Fix_{0-1000m}$ | $N_2Fix_{SML}/N_2Fix_{euphotic}$ | $N_2Fix_{euphotic}/N_2Fix_{0-}$ |
|---|---|---|---|---|---|---|
| | $\mu molN\ m^{-2}\ d^{-1}$ | $\mu molN\ m^{-2}\ d^{-1}$ | $\mu molN\ m^{-2}\ d^{-1}$ | $\mu molN\ m^{-2}\ d^{-1}$ | % | % |
| ST01 | 14.6 | 42.6 | 56.5 | 99.1 | 34 | 43 |
| ST02 | 10.7 | 36.0 | 16.0 | 51.9 | 30 | 69 |
| ST03 | 7.8 | 58.3 | 18.1 | 76.4 | 13 | 76 |
| ST04 | 10.8 | 46.6 | 38.5 | 85.1 | 23 | 55 |
| ST05 | 4.9 | 46.3 | 36.1 | 82.4 | 10 | 56 |
| TYR | 4.2 | 38.6 | 53.0 | 91.6 | 11 | 42 |
| ST06 | 9.1 | 34.9 | 29.8 | 64.7 | 26 | 54 |
| ST07 | 10.5 | 43.5 | 55.4 | 98.8 | 24 | 44 |
| ION | 6.2 | 40.6 | 56.5 | 97.1 | 15 | 42 |
| ST08 | 4.3 | 27.0 | 12.3 | 39.3 | 16 | 69 |
| ST09 | 3.4 | 50.2 | 43.3 | 93.5 | 7 | 54 |
| FAST | 5.9 | 58.2 | 35.7 | 93.8 | 10 | 62 |
| ST10 | 13.7 | 1908 | 63.7 | 1972 | 1 | 97 |
| Mean ± std (ST10 excluded) | 7.7±3.5 | 44±9 | 38±16 | 81±20 | 18%±9% | 55%±12% |
| Mean ± std (all stations) | 8.2±3.7 | 187±517 | 40±17 | 227±525 | 17%±10% | 59%±16% |

- Table 2: Please specify what size fraction (or incubation experiment) is used to compute the C:N (mol/mol) ratio?

Reply to RC1: the C:N ratio in Table 2 corresponds to the POC:PN ratio calculated from the IRMS measurements of the GFF filters ($> 0.7\mu m$).

We have added in the legend of Table 2 (revised version) (in bold) 'Initial physico-chemical and biological properties of surface seawater before the perturbation in the dust seeding experiments at TYR, ION and FAST (average at T0 in C and D treatments, n=4 or data at T-12h in the pumped surface waters, n=1). The relative abundances of diazotrophic cyanobacteria and NCD (non-cyanobacterial diazotroph) are given as proportion of total *nifH* sequence reads. DIP: dissolved inorganic phosphorus, DFe: dissolved iron. **The C:N ratio corresponds to the ratio in the organic particulate matter from IRMS measurements ($> 0.7\mu m$).** Means that did not differ significantly between the experiments ($p>0.05$) are labeled with the same letter (in parenthesis). '

- Figure 1: Station "TYR" labelled as "TYRR" : This was changed in Fig. 1

- Figure 2: Are data points missing at 1000 m for ST6, ST8 and ST10? Authors should consider breaking the scale of the x-axis ($N_2$ fixation, nmol N $L^{-1}$ $d^{-1}$) for station 10. This would improve the readability of the graph, and highlight significant $N_2$ fixation rates, not only at 61 m.

For the sake of clarity, we have symbolized by crosses the N2 fixation rates under detection limit (<0.04 nmol N $L^{-1}$ $d^{-1}$) on Fig. 2. $N_2$ fixation rates at station 10 are now plotted in log scale to improve the readability of the figure

[Figure]

- Figure 6 Please adjust the y-axis to a unique range for all 3 graphs and arrange the graphs side by side.

This has been modified in the revised version

- Figure S5: Please consider dissociating the stations either into separate plots, or even just separated series on the same plot.

This figure has been modified in the revised version (see below)

[Figure]

Figure S5: **Changes in the general diversity trends visualized by Shannon H index, during the dust seeding experiments at TYR, ION and FAST between initial time (dot) and final time (square) connected by a line to indicate directional change in diversity following each incubation experiment.** Shows that for TYR and ION the diversity decrease from T0 to Tend whereas the opposite is true for FAST

**TECHNICAL CORRECTIONS:**

Lines 24-25: "strong longitudinal gradient increasing eastward" → **corrected**

Line 72: "enhance" instead of "enhanced" → **corrected**

Line146: space missing between "and" and "$^{13}$C" → **corrected**

Line 153: Adapt reference "Nagata, 1986" instead of "Nagata et al., 1986" → **replaced by Fukada et al., 1998**

Line 158: replace "following" by "as follows"→ **corrected**

Line 159: For the sake of clarity, the variable Tx could be removed from the formula, since the term cancels itself being in both the numerator and denominator. On the other hand, "$N_2FIXATION_T$" could be replaced by "$N_2FIXATION_{Tx}$"→ **corrected**

Line 244: replace "as" by "or"→ **corrected**

Line 318: Add in the parentheses "(**in** this and previously published studies)".→ **corrected**

Line 337: delete "and", to read "… take place, combined with…"→ **corrected**

replace "high stocks" by "higher stocks"→ **corrected**

Line 349: "whole diazotrophic community in the euphotic zone" instead of "the whole diazotrophs"→ **corrected**

Reference Table S1 at the end of the sentence → **added**

Line 384: Data reported here do not support an increase of diazotrophs abundances, so consider replacing "obviously" by "likely".→ **completely agree**

Line 398: replace "**to** dust seeding" by "**by** dust seeding → **corrected**

Line 406: please clarify, "heterotrophic prokaryotes, NCD, and photoautotrophs" had to compete for dust-derived DIP

Revised to **'Consequently, diazotrophs as well as non diazotrophs (heterotrophic prokaryotes and photoautotrophs) could all uptake the dust-derived DIP reducing then potentially the amount of DIP available for each cell that could explain the lower stimulation of $N_2$ fixation relative to TYR'**

Line 407: "the lower stimulation" instead of "the lowest"→ **corrected**
Line 477: "UCYN-A remain" instead of "remained"→ **corrected**
Line 480: "are expected" instead of "would expect"→ **corrected**

References added in the revised version:

Berman-Frank, I.A., Quigg, A., Finkel, Z. V, Irwin, A.J., Haramaty, L. (2007) Nitrogen-fixation strategies and Fe requirements in cyanobacteria. Limnol. Oceanogr. 52, 2260–2269.

Fukuda, R., Ogawa, H., Nagata, T., & Koike, I. (1998). Direct determination of carbon and nitrogen contents of natural bacterial assemblages in marine environments. Applied and Environmental Microbiology, 64(9), 3352–3358. https://doi.org/10.1128/aem.64.9.3352-3358.1998

Tuit, C., Waterbury, J., Ravizza, G. (2004) Diel variation of molybdenum and iron in marine diazotrophic cyanobacteria. Limnol. Oceanogr. 49, 978–990. doi:10.4319/lo.2004.49.4.0978

---

## Author Comment (AC2)

We appreciate the reviewers' interest and thank them for their relevant suggestions and time spent reviewing this manuscript

**REPLY to referee RC2**

**General comments**

Hence, overall the quality of the manuscript is high and it makes an insightful contribution to our understanding of diazotrophs and their activity in the Mediterranean Sea. In saying that, I do still have some questions about the data interpretation and in particular, the reliance of one key outcome of the paper on a high $N_2$ fixation rate measured at 1 station and at 1 depth which is ~x100 higher than any other measured (volumetric) $N_2$ fixation rate. The dedicated section 4.3 to "Intriguing station 10" aims to explain this observation citing studies with similar magnitude rates and suggesting that this is due to the patchiness often observed with UCYN-A abundances, the dominant diazotroph detected. The authors argue that this is likely due to nutrient inputs from Atlantic water intrusion into the surface and a different diazotrophic community present.

**Q1 : Were replicate incubations made for each sampled depth to indicate if this is a reproducible result? If yes, it would be helpful to report the standard deviation of the rates to indicate variability in the measurements. If not, then I would question how robust this finding is.**

Reply to RC2: One sample per depth was collected on the TMC rosette; this was added in MM in the revised version; N2 fixation is a parameter that requires a large volume of water and unfortunately there was not enough water in the Go-Flo bottles after the samples collection from all the cruise participants.

As mentioned in the manuscript, the N2 fixation rate at the DCM at station 10 (72 nmol N L-1 d-1) is very high compared to those measured at other stations/depths. The rate at 37m is also high ~ 3 nmol N L-1 d-1. We are confident that this high value is robust for the following reasons:

-Simultaneous data of particulate C and isotopic 13C ratio measured on the same GFF filter than particulate N and isotopic 15N ratio are consistent
-Particulate N (PN) and 15N isotopic ratio of certified reference materials measured before and after this sample are consistent
-15N and PN in PEACETIME spiked samples measured before and after this sample are consistent
-GFF blanks are negligible
-PN and 15N measured on the natural seawater at this depth (61m) are consistent

**Q2 One other limitation of the presented data set that is also acknowledged by the authors, is that no quantitative nitrogenase gene analysis was carried out and all conclusions are based on qualitative data on community composition.**

We fully agree with the reviewer; we mentioned this gap in the discussion of the submitted version. Quantitative data would have allowed us to validate some of our assumptions,

allowing to go further in the interpretations. Unfortunately we did not perform qPCR on our samples.

I would also encourage the authors to make the data openly available, latest at publication, rather than keeping it embargoed until 2023.

The PEACETIME data set will be available on SEANOE (https://doi.org/10.17882/75747) upon publication of all MS in the special issue (December 2021)

**Specific comments**

- Line 31: The Mediterranean Sea is generally considered a desert because of very low surface nutrient concentrations so it is puzzling to see "nutrient rich" here used to describe some stations, as the measured surface concentrations were in the nanomolar range.

**revised to (in bold):** 'These in situ observations of greater relative abundance of UCYN-A at stations with higher nutrient concentrations and dominance of NCD at more oligotrophic stations suggest that nutrient conditions **- even in the nanomolar range -**may determine the composition of diazotrophic communities and in turn N2 fixation rates.'

- Line 37: It isn't clear how $N_2$ fixation could be "exacerbated". Consider rephrasing to "increased" or similar.

This was changed in 'Under projected future conditions, $N_2$ fixation was either **increased** or unchanged'

- Lines 75-77: The statement that atmospheric inputs would be particularly important for diazotrophic organisms under increased stratification due to ocean warming is not well explained. Why would diazotrophic organisms in particular be affected?

The sentence was confusing; we have rewritten it in 'Future sea surface warming and associated increase in stratification (Somot et al., 2008) might reinforce the importance of atmospheric inputs as a source of new nutrients for biological activities during that season, **including** diazotrophic microorganisms.

- Line 101: Was the metabolic activity of diazotrophs present measured at this station or was this rather referring to anticipated differences in metabolic activity due to differences in oligotrophic conditions?

The sentence was changed in 'Based on previous studies, the location of the three long stations was chosen based on several criteria including because they represent three main bioregions of the MS (Guieu et al., 2020, their Fig. S1). They are located along the longitudinal gradient in biological activity, including the activity of diazotrophs decreasing eastward (Bonnet et al., 2011; Rahav et al., 2013a)'

- Line 104: Unfiltered seawater was used for the incubations. Does this mean that larger grazers could have been present and influenced the biomass development or nutrient regeneration inside the incubations?

Yes grazers were present in the seawater used for the dust seeding experiments; the impact of grazing (including from the larger ones) could of course have influenced phytoplanktonic biomass through a potential top-down control; this is described in the companion paper of Gazeau et al., (2021a) which also presents the data on the abundances of meso-zooplankton species at the end of the experiments (their figure 9).

- Lines 116 – 122: A figure or table as an overview of all key steps in setting up the dust incubation experiments from dust preparation, to $CO_2$/temperature manipulation and final sampling would be helpful.

Reply to RC2: Chemical and mineralogical features of the dust used in the dust seedings experiments are fully described in Guieu et al. (2010b) as well as the protocol of cloud processing. Moreover, as mentioned in MM, the experimental setup of the dust seeding experiments is fully described in the published companion paper of Gazeau et al., (2021a) including $CO_2$/temperature manipulation and final sampling. We decided not to add in our paper these data already published in order not to weigh down our manuscript. **The succession of operations is fully described in Gazeau et al. (2021a, see their Table 1).** This last sentence was added in the revised ms.

- Line 125: Concentration of HCl used for acid washing is missing.

Revised version: 'All materials were acid washed (HCl Suprapur **32%**) following trace metal clean procedures.'

- Section 2.3: Were blank incubations (i.e. without isotope addition) carried out to correct for any incubation effects?

Reply to RC2: Absolutely. 2.3 L of seawater without 15N and 13C additions were filtered onto precombusted GFF filters to determine natural concentrations and isotopic signatures of particulate carbon and nitrogen.

- Line 137: The incubation irradiances are reported as "percentages of attenuation", however it seems this might be more accurately reported as "transmittance"? The order of the values from highest to lowest would indicate the lowest irradiance first (70% attenuation). Is this correct? What type of blue filter was used? Also, to what depths do these attenuations correspond to?

Reply to RC2: As our samples were incubated on the same conditions of light than samples for 14C-PP presented in the companion paper of Maranon et al., 2021, we have chosen for the sake of clarity between the 2 articles, to express the % irradiance as in Maranon's paper. For more clarity, we have added some additional information

We have added in MM in the revised version (in bold) 'The *in situ* samples from the euphotic zone were incubated in on-deck containers with circulating seawater, equipped **with different sets of blue neutral density filters (Lee Filters)** (percentages of attenuation: 70, 52, 38, 25, 14, 7, 4, 2 and 1 %) to simulate **an irradiance level (% PAR) as close as possible to the one corresponding to their depth of origin'**

- Line 150: Here it states the "molar C:N ratio in the particulate matter was calculated and used to estimate the contribution of $N_2$ fixation to primary

production". How was this exactly done? What impact might detritus have on this calculation? How does this compare to the N demand as calculated from the measured PP rates rather than POC concentrations?

Reply to RC2: We have converted N2 fixation rate in carbon using the molar C:N ratio measured in the particulate matter in order to obtain a rate in nmolC L-1 d-1. Then we compared this rate to the primary production and expressed it as a %. We showed that over the peacetime cruise $N_2$ fixation was a poor contributor to PP ($1.0 \pm 0.3$ % of PP) but that this process could supply up to 20 % of the bioavailable N requirement to support PP at station 10.
The C:N ratio in the particulate matter (> 0.7 µm, GFF filter) measured during the cruise indeed includes detrical material; nevertheless detrical material in the euphotic layer having a C:N ratio close to that of phytoplankton (Schneider, 2002), our conversion is accurate.

- Line 233: Here the surface is specifically mentioned as 5m deep, but this distinction isn't clear in other instances e.g. Fig. 3 and the surface mixed layer is also used to define the surface layer for reporting integrated rates and stocks. This is a little confusing and the changing definition isn't justified in the text for each variable, which makes it difficult as a reviewer to accurately scrutinise and assess the results. If different definitions are necessary to highlight key relationships between variables in different water column section, this should be stated and justified more clearly. If atmospheric deposition is a key nutrient input that drives observed variability in diazotroph activity and community composition, I would imagine the surface mixed layer would be the best definition to be used, rather than just 5m. Table S1 suggests the SML was indeed deeper than 5m and up to 21 meters deep. Following from this, I was a little confused by the observation that surface $N_2$ fixation rates (5m) and euphotic zone but NOT the surface mixed layer nor aphotic $N_2$ fixation rates correlated with longitude. As far as I could tell, this wasn't picked up in the discussion at all but would be interested to understand this result better. What could be possible explanations for this observation?

Reply to RC2: Volumetric rate of N2 fixation measured at ~5m depth exhibited a longitudinal gradient decreasing eastward (r = -0.61 and r = -0.60, p < 0.05, respectively) (Fig.3) while $N_2$ fixation rates integrated over the SML (defined as the layer from 0m to the mixed layer depth (MLD) ranging from 7m to 21m, Table S1) displayed no significant trend with longitude (p > 0.05). In addition, we chose to present the longitudinal gradient of the volumetric N2 fixation at the surface because we also measured at some stations the relative composition of the diazotrophic communities at the same depth (cf discussion section 4.2).

We clarified this in the revised version in MM as: **'Volumetric surface** (~ 5 m) and euphotic layer integrated $N_2$ fixation rates exhibited a longitudinal gradient decreasing eastward (r = -0.61 and r = -0.60, p < 0.05, respectively) (Fig.3). Integrated $N_2$ fixation rates over the SML **(Table S1)**, aphotic and 0-1000 m layers displayed no significant trend with longitude (p > 0.05).'
and in Figure 3: '**Volumetric** surface **(~5m)** (a) and integrated $N_2$ fixation from surface to euphotic layer depth (b) along the longitudinal PEACETIME transect (station 10 was excluded).'

In agreement with our results, Benavides et al., (2016) didn't find a longitudinal gradient of aphotic N2 fixation. The fact that there is no correlation between longitude and aphotic N2 fixation while there is one with euphotic N2 fixation, could mean that aphotic and euphotic N2 fixation are controlled by different limiting factors .

We were also surprised to find no correlation between longitude and integrated N2 fixation over the SML while we did find one over the euphotic layer. This could be due to the fact that the depth of the SML varied a lot (factor of ~3; coefficient of variation =35%) while the variation of the euphotic layer depth was low (+/- 9m; CV = 12%) at the PEACETIME stations. Furthermore, there is a strong positive correlation between the MLD and integrated N2 fixation over the SML (r=0.86, p<0.01) while there is no correlation between the depth of the euphotic layer and N2 fixation integrated over the euphotic layer

- Section 2.6: It isn't clear how the missing nanomolar nutrient concentrations at Station 1-4 were taken into account in the statistical analyses and if this may have an influence on the correlation analysis output. Table S1 does indicate that a maximum concentration of 0.05 µmol L$^{-1}$ was used when calculating the $NO_3^-$ stocks. Was the same approach used for the Pearson correlation? If yes, how may this have affected any potential correlations for the surface mixed layer or where depths <50m were included in the calculation?

You are totally right. We forgot to mention it. To test the potential correlation between DIN stocks and integrated N2 fixation, PP and BP, the estimated DIN stocks at stations 1-4 were not taken into account (n=8 or 9).

We have added in the revised version in section 2.6 (in bold): 'Pearson's correlation coefficient was used to test the statistical linear relationship (p < 0.05) between $N_2$ fixation and other variables (BP, PP, DFe, DIP, $NO_3^-$); **it should be noted that the DIN stocks estimated at stations 1 to 4 (Table S1) were excluded from statistical analysis'**

- Line 253: The exclusion of the $N_2$ fixation rates from Station 10 can be appreciated due to the one depth that has remarkably high rates but this does lack clear justification in the manuscript. Please also see further comments on this one station below.

We have added in the revised version in section 2.6 (in bold) 'For statistical analysis, **due to the high integrated $N_2$ fixation rate from station 10,** this rate was not included in order not to bias the analysis.'

- Line 307-309: The detection of UCYN-A3 and -A4 sublineages is an exciting new discovery for the region. Is there a particular reason why these groups were now detected? Is this due to methodological developments or rather due to the oceanographic conditions present?

We believe that it is primarily the database. The UCYN-A compilation made by Farnelid et al (2016) and the oligotyping database compiled by Turk-Kubo et al (2017) has not been applied to amplicon data in this area before

- Line 340: Why is this low DFe not explained solely by diazotroph uptake? As no quantitative data is reported on abundances, this is difficult to assess.

We estimated the theoretical Fe requirement to sustain a N2 fixation of 72 nmol N L$^{-1}$ d$^{-1}$ at 61m, station 10 using a range (min-max) of Fe/C (from 7 to 177 µmol:mol) and associated C/N for diazotrophs  (*Trichodesmium*, UCYN) from literature (Berman-Frank et al., 2007; Tuit et al., 2004, Jiang et al., 2018). We found that to sustain this N$_2$ fixation rate, 0.004 nM to 0.08 nM of DFe are required. Consequently, the minimum in DFe concentration at 61m of 0.47 nM compared to 0.7 to 1.4 nM at the nearby depths, (Bressac et al., 2021)  could not be explained solely by the diazotrophs uptake.

We have added in the revised version (in bold): 'It only coincided with a minimum in DFe concentration (0.47 nM compared to 0.7 to 1.4 'nM at the nearby depths, Bressac et al., 2021). **Based on a range of Fe:C (from 7 to 177 µmol:mol) and associated C:N ratios for diazotrophs (*Trichodesmium*, UCYN) from literature (Berman-Frank et al., 2007; Tuit et al., 2004; Jiang et al., 2018 ), we found that 0.004 nM to 0.08 nM of DFe are required to sustain this N$_2$ fixation rate. Consequently, the minimum in DFe concentration at 61m could not be explained solely by the diazotroph uptake.'**

- Line 398: What is considered the limiting factor for N$_2$ fixation at TYR and ION that was not considered limiting at FAST? Final rates in the dust incubations were actually quite similar between the three stations but the difference in trends in % difference in rates appears to be driven by the different baseline at the different stations e.g. the baseline at FAST is higher (~0.5 vs ~0.2 nM N L$^{-1}$ d$^{-1}$). Could this mean the diazotrophs at all stations have the same potential to fix N but are just limited under ambient conditions (without dust/nutrient inputs). It seems like the ION community are only nutrient limited, yet in TYR and FAST are below their thermal optimum conditions. The idea of temperature optima is brought up in regards shifts in the diazotrophic community within a station (lines 450-454) but could this also be important between the three studied regions of the Mediterranean Sea?

Based on the responses of N2 fixation, PP and BP after dust seedings, our results suggest that NCD-supported N2 fixation is not limited by organic C at TYR while it might be at ION and FAST; DIP might also be a limiting or co-limiting factor for N2 fixation at all stations.

The relative change (%) in N2 fixation after dust seeding is not solely driven by the baseline as it is similar at TYR and ION while the relative change is twice as high at TYR (+321%) than at ION (161%) (L387-388 and L399). As shown by the difference in the initial N2 fixation rates (Table 2), N2 fixation was initially more limited at TYR and ION compared to FAST (L397). It is unlikely that diazotrophs at all stations have the same potential to fix N2 as UCYNA is the dominant diazotroph at FAST whereas it is NCD at TYR and ION, and the cell-specific N$_2$ fixation rates have been shown to be higher for UCYN-A relative to NCD (Turk-Kubo et al., 2014; Bentzon-Tilia et al., 2015; Martinez-Perez et al., 2016; Pearl et al., 2018; Mills et al., 2020) (L322-323).

We believe that the small difference in temperature between the 3 stations does not explain the differences in the biological responses for the following reasons. Initial difference T° between the three stations is low (maximum ΔT°~1°C between TYR and FAST), and there is no correlation between temperature and volumetric N2 fixation rate (p=0.71) measured during the cruise. TYR and ION were both dominated by NCD and characterized by the same initial N2 fixation rate. The temperature at ION was 0.6°C higher than at TYR, so

such a difference in T° does not seem to influence the relative composition of the diazotrophic community and its N2 fixation rate. FAST, dominated by UCYNA, is characterized by a 2.5-fold higher N2 fixation rate than at ION dominated by NCD, whereas the temperature was only 0.3°C higher at FAST than at ION. Furthermore, the temperature range in temperate regions where UCYNA dominates, appears to be wide, from ~**20°C** to 24° (Langlois et al., 2008; Moisander et al., 2010).

- Lines 406-408: The final sentence in this paragraph was confusing to me, in particular how the three group (heterotrophic prokaryotes, NCD, photoautotrophs) would outcompete and thereby reduce the DIP taken up by each cell. I'm not sure "outcompete" is the right word here and would recommend rephrasing this sentence in a more simple manner.

The sentence was replaced by 'Consequently, diazotrophs as well as non diazotrophs (heterotrophic prokaryotes and photoautotrophs) could all uptake the dust-derived DIP reducing then potentially the amount of DIP available for each cell that could explain the lower stimulation of $N_2$ fixation relative to TYR'

- Lines 409-412: Could the symbiosis be stimulated the other way around i.e. dust enhances $N_2$ fixation in UCYN-A which then relieves N-limitation in the photoautotrophic host? This would fit better with the "potential N limitation" and nutrient ratios reported e.g. lines 264-267.

Good point, we have added in the revised ms in section 4.5 (in bold) 'At FAST, initially dominated by UCYN-A, $N_2$ fixation and PP correlated strongly after the dust seeding (**Fig S8c**). This indicated that dust could relieve either directly the ambient nutrient limitation of both $N_2$ fixation and PP (Fig.S3) or indirectly through first the relief of the PP limitation of the UCYN-A photoautotroph hosts inducing an increase in the production of organic carbon which could be used by UCYN-A to increase its $N_2$-fixing activity. **Nutrients from dust could also first enhance the UCYNA-supported N2 fixation, which in turn could relieve the N limitation of the UCYN-A photoautotrophic host, as the initial $NO_3^-$/DIP ratio indicates a potential N limitation of the PP (Table 2).'**

- Lines 419-423: It is true that changes in $CO_2$ concentrations would not directly affect diazotrophs that do not fix $CO_2$ however there are associated changes in seawater pH under ocean acidification (OA) that may affect cellular metabolism on short time scales in both autotrophic and non-autotrophic organisms. This, probably more relevant, change in seawater chemistry for non-autotrophs is not acknowledged here in this introductory paragraph although mentioned later in lines 441-443. I would suggest pH should be more clearly stated as the key factor rather than $CO_2$, even if observations indicate OA does not seem to affect communities dominated by UCYN-A.

We have added in the revised ms section 4.6 (in bold) 'The purpose of our study was to study the combined effect of warming and acidification, but we can expect on the short time scale of our experiments (< 3-4 days), that NCD and UCYN-A would not be directly affected by  the  changes in the $CO_2$ concentration as they do not fix $CO_2$ (Zehr et al., 2008). Indeed, no impact of acidification (or $pCO_2$ increase) on $N_2$ fixation was detected when the diazotrophic communities were dominated by UCYN-A in the North and South Pacific (Law et al., 2012; Böttjer et al. 2014). **Nevertheless, the**

**decrease in pH may indirectly impact UCYN-A through changes affecting its autotrophic host.'**

- Line 453-458: The shift to larger diazotrophs with a higher cell-specific $N_2$ fixation rate under ocean warming could explain the stimulation in $N_2$ fixation rates but is difficult to determine from relative nifH-based community composition. I feel the that here, care needs to be taken to not to confuse increase in absolute abundance (as observed in the cited study by Henke et al. 2018 for UCYN-A2) with an increase in the relative abundance which was what was measured in this study. Any increase in $N_2$ fixation would still need sufficient resources (e.g. nutrients) to fuel this unless there was an underlying change in organism metabolism with temperature/pH. Differences in community composition may drive changes in observed activity but differences in abundances would arguably have a larger impact on measured rates. Furthermore, could the increase in N provided by stimulated N-fixation by NCDs not account for the increase in PP also observed for the FAST station? As NCDs were the dominant diazotroph detected, this, to me at least, would make more sense.

As mentioned in section 4.2, unfortunately we do not have quantitative abundances of diazotrophs in this study. This is why we remain very cautious, preferring only suggesting hypotheses to explain some of our results as it is the case here.

As there is no correlation between N2 fixation and BP after dust seeding at FAST (see below Fig S4 in the revised ms), NCD are probably not responsible for the increased N2 fixation. Rather, UCYN-A in association with prymnesiophytes could be responsible for the majority of the enhanced $N_2$ fixation as $N_2$ fixation correlated strongly with PP (section 4.5) (see Fig. S4 below).

Revised to: 'Interestingly, Henke et al. (2018) observed that the **absolute** UCYN-A2 abundance was positively affected by increasing temperature, within a range of temperature from about 21 to 28° C which is in agreement with our results **although only relative abundances were measured in our study**. Based on the strong positive correlation between $N_2$ fixation and PP after dust addition **(and no correlation between $N_2$ fixation and BP, Fig S4)**, and despite the decrease in the relative abundance of UCYN-A3, the increased stimulation of $N_2$ fixation under future climate conditions could **likely** be sustained by the increase in the relative abundance of UCYN-A2 which is bigger than UCYN-A3 (Cornejo-Castillo et al., 2019) and could consequently have a higher cell-specific $N_2$ fixation rate'

We have added this figure in SI as also suggested by the reviewer RC1

[Figure]

**Figure S4:** Relationship between N₂ fixation and (a) bacterial production (BP) at TYR, (b) BP at ION, and (c) primary production (PP) at FAST, during the dust seeding experiments

Pearson correlation coefficient (and associated p-value, in bold p-value<0.05) between N₂ fixation rates and bacterial production, primary production (¹³C), measured during the dust seeding experiments at stations TYR, ION and FAST

|  | N₂ Fix-TYR | N₂ Fix-ION | N₂ Fix-FAST |
|---|---|---|---|
| BP | 0.82 (<0.0001) | 0.76 (0.001) | 0.36 (0.17) |
| ¹³PP | 0.59 (0.12) | 0.44 (0.09) | 0.90 (<0.0001) |

- Line 469: The phrase "triggered N₂ fixation" implies that there was no N₂ fixation happening before. Instead, this is probably referring to the observed increase in N₂ fixation in the dust/warming/acidification incubations. Please consider rephrasing.

We have added in the revised version (in bold): 'Through the release of new nutrients, simulated wet dust deposition under present and future climate conditions significantly **stimulated** N₂ fixation'

- Figure 2: It is intriguing that the N₂ fixation rates were highest below the surface rather than in the top 5 m considering the importance of atmospheric deposition. Is this related to other processes such as diffusive nutrient supply? It might be helpful to indicate the nutrient depth profiles, perhaps in the supplementary material, to explain this.

Within the SML, N2 fixation rates are often higher at 5m which could be related to new P from atmospheric inputs. Within the euphotic layer, N2 fixation rates are often maximum close to the DCM as also observed by Benavides et al., 2016 and Bonnet et al., 2011. As shown in the companion paper of Maranon et al. (2021), the depths of both the nitracline and the phosphacline were strongly correlated with the DCM depth throughout the PEACETIME cruise ($p < 0.001$). So, the fact that the N2 fixation is relatively high close to the DCM could be related to the vertical diffusive inputs of DIP and to the supply of organic carbon from phytoplanktonic activity for NCD.

- Figure 6: It would be helpful if the x-axis had the same scale for each site i.e. proportionally longer for FAST as the incubation lasted 4 days instead of 3. Also, why were some days excluded from the analyses? These seem to be robust rate estimates. Was only the nutrient replete period of interest here? It would be useful to know why a linear relationship was expected by the authors that lead them to a linear regression analysis rather than an approach such as mixed models.

This has been changed in the revised version on Fig. 6; the x-axis now has the same scale (0 to 4 days). To test significant differences between the slopes of $N_2$ fixation as a function of time in the C, D and G treatments, we used the data presenting a significant linear relationship with time. So, we have to exclude the data at T3 days at ION (D and G treatments) and at TYR (G treatment). We used a simple linear model because the vast majority of our data showed a linear relationship with time.

- Figure S5: The use of a line plot connecting the Shannon H index is a little misleading as it suggests all data are connected along the x-axis and I would recommend revising this figure. As there are multiple stations included in this figure, I would recommend breaking the line between stations to highlight the change in diversity over time which is, what I understand, the most important aspect here but is not clearly portrayed in the current figure. Different symbols for the different stations would also be helpful to demonstrate the increase in diversity for FAST vs. decrease in diversity over time for TYR and ION. An x-axis label would also be helpful.

This figure has been modified in the revised version (see below)

[Figure]

Figure S5: Figure S5: **Changes in the general diversity trends visualized by Shannon H index, during the dust seeding experiments at TYR, ION and FAST between initial time (dot) and final time (square) connected by a line to indicate directional change in**

**diversity following each incubation experiment.** Shows that for TYR and ION the diversity decrease from T0 to Tend whereas the opposite is true for FAST

- Table 1: Consider reporting the standard deviation for the three distinct areas to further highlight

As one sample per depth was collected to determine N2 fixation rate, we are not able to calculate a mean and standard deviation

- In general, addition of important information to panels such as the station (e.g. Fig. 8) would also make it easier for the reader to grasp the figure, rather than keeping it in the figure captions. A table with an overview of the Pearson correlation coefficients would be a useful addition.

We have added on Figure 8 'TYR, ION, FAST' next to the graphs. We have also added in Fig S4 a table with the Pearson correlation coefficient and associated p-value, between $N_2$ fixation rates and BP, and PP measured during the dust seeding experiments.

Pearson correlation coefficient (and associated p-value, in bold p-value<0.05) between $N_2$ fixation rates and bacterial production, primary production ($^{13}C$), measured during the dust seeding experiments at stations TYR, ION and FAST

|  | $N_2$ Fix-TYR | $N_2$ Fix-ION | $N_2$ Fix-FAST |
|---|---|---|---|
| BP | 0.82 **(<0.0001)** | 0.76 **(0.001)** | 0.36 (0.17) |
| $^{13}PP$ | 0.59 (0.12) | 0.44 (0.09) | **0.90 (<0.0001)** |

**Technical corrections**

Line 79: "nutrients repleted" should be "nutrient repleted". **→ corrected**

Line 84: "nutrients" should be "nutrient" and "diazotrophic communities" should be "diazotrophic community". **→ corrected**

Line 168: Individual stations should have a capital S i.e. "Station 10" or "Stations 1 and 10" on Line 226, whereas "stations", when referring in general (e.g. Line 255) do not need capitalisation. **→ corrected**

Line 217: Please consider adding citations to specific R packages used within the software to acknowledge the package authors.

revised to: 'Statistical tests were done using XLSTAT and R (version 4.1.1) with the stats, tidyverse, FactoMineR packages.'

Line 336: "exchanges" should be "exchange".**→ corrected**

Line 340: "diazotrophs uptake" should be "uptake by diazotrophs" or "diazotroph uptake".**→ corrected**

---

## Author Response (AR2)

We thank the Co-Editor-in-Chief, Christine Klaas, for all her suggestions that clearly improve this manuscript. The two technical corrections asked by the Associate Editor Carolin Löscher have all been taken into account.

Best regards,

Céline Ridame on behalf the co-authors.